# LONG-TERM FAIRNESS IN REINFORCEMENT LEARNING WITH BISIMULATION METRICS

## ABSTRACT

Ensuring long-term fairness is crucial when developing automated decision making systems, specifically in dynamic and sequential environments. By maximizing their reward without consideration of fairness, AI agents can introduce disparities in their treatment of groups or individuals. In this paper, we establish the connection between bisimulation metrics and group fairness in reinforcement learning. We propose a novel approach that leverages bisimulation metrics to learn reward functions and observation dynamics, ensuring that learners treat groups fairly while reflecting the original problem. We demonstrate the effectiveness of our method in addressing disparities in sequential decision making problems through empirical evaluation on a standard fairness benchmark consisting of lending and college admission scenarios.

## 1 INTRODUCTION

As machine learning continues to shape decision making systems, understanding and addressing its potential risks and biases becomes increasingly imperative. This concern is especially pronounced in sequential decision making, where neglecting algorithmic fairness can create a self-reinforcing cycle that amplifies existing disparities (Jabbari et al., 2017; D'Amour et al., 2020). In response, there is a growing recognition of the importance of leveraging reinforcement learning (RL) to tackle decision making problems that have traditionally been approached through supervised learning paradigms, in order to achieve long-term fairness (Nashed et al., 2023). Yin et al. (2023) define long-term fairness in RL as the optimization of the cumulative reward subject to a constraint on the cumulative utility, reflecting fairness over a time horizon.

Recent efforts to achieve fairness in RL have primarily relied on metrics adopted from supervised learning, such as demographic parity (Dwork et al., 2012) or equality of opportunity (Hardt et al., 2016b). These metrics are typically integrated into a constrained Markov decision process (MDP) framework to learn a policy that adheres to the criterion (Wen et al., 2021; Yin et al., 2023; Satija et al., 2023; Hu & Zhang, 2022). However, this approach is limited by its requirement for complex constrained optimization, which can introduce additional complexity and hyperparameters into the underlying RL algorithm. Moreover, these methods make the implicit assumption that stakeholders are incorporating these fairness constraints into their decision making process. However, in reality, this may not occur due to various external and uncontrollable factors (Kusner & Loftus, 2020).

In this work, we highlight a surprising connection between group fairness in RL and the bisimulation metric (Ferns et al., 2004; 2011), an equivalence metric that captures the behavioral similarity between states. We show that minimizing the bisimulation metric between members of different groups results in demographic parity fairness. Building upon this insight, we propose a practical algorithm that, guided by the bisimulation metric, adjusts the reward and observation dynamics (how the observations change in the environment) to achieve long-term fairness in RL.

By modifying the observable MDP—the rewards and the observations seen by the agent—we show that unconstrained policy optimization inherently satisfies the fairness constraint in the original, unmodified MDP. This concept is analogous to real-world practices, where regulatory frameworks are established to influence decision making processes—for instance, governments impose lending regulations on banks to ensure fairness and equity (FDIC, 2005). A significant advantage of our method is that it does not require modifying the underlying RL solver. This allows us to lever-

age the strengths of deep RL while avoiding the complexities and intricacies associated with other constrained optimization methods used to achieve fairness in RL.

Through comprehensive evaluation on a standard fairness benchmark (D'Amour et al., 2020), widely used in the literature (Xu et al., 2024; Deng et al., 2024; Hu et al., 2023; Yu et al., 2022), we show that our unconstrained approach outperforms strong baselines for long-term fairness. Our code is submitted in the supplemental material and will be publicly available. Our contributions are:

1. Establishing the connection between bisimulation metrics and group fairness in RL.
2. Developing a novel method that allows unconstrained optimization of a policy to automatically achieve demographic parity fairness.
3. Implementing a practical algorithm, guided by bisimulation metrics, that when coupled with an unmodified RL algorithm, achieves fairness on a standard benchmark.

Ultimately, the connection to bisimulation metrics offers a novel unconstrained perspective on achieving fairness in RL, and we establish the initial foundations in this direction.

## 2 BACKGROUND

We consider an MDP defined by a 5-tuple $(\mathcal{S}, \mathcal{A}, \tau_a, R, \gamma)$, with *state space* $\mathcal{S}$, *action space* $\mathcal{A}$, *transition dynamics* $\tau_a : \mathcal{S} \times \mathcal{A} \to \text{Dist}(\mathcal{S})$, where $\text{Dist}(\mathcal{S})$ is the probability simplex over $\mathcal{S}$, *reward function* $R : \mathcal{S} \times \mathcal{A} \to \mathbb{R}$, and *discount factor* $\gamma \in (0, 1]$. The *Value function* $V^\pi(s_t) = \mathbb{E}_\pi \left[ \sum_{k=0}^\infty \gamma^k R(S_{t+k}, A_{t+k}) \mid S_t = s \right]$ denotes the expected return from $s$ under policy $\pi$. The goal is to find a policy $\pi : \mathcal{S} \to \text{Dist}(\mathcal{A})$ that maximizes the expected return $J^\pi = \mathbb{E}_{s \sim \rho^\pi(s)}[V^\pi(s)]$.

The *bisimulation relation* for MDPs (Desharnais et al., 2002; Givan et al., 2003) captures the concept of behavioral similarity and is defined below.

**Definition 1** (Bisimulation). A *bisimulation relation* on an MDP $\mathcal{M}$ is an equivalence relation $B \subseteq \mathcal{S} \times \mathcal{S}$ such that if $s_i B s_j$ holds for $s_i, s_j \in \mathcal{S}$, the following properties are true:

$$R(s_i, a) = R(s_j, a) \quad \text{and} \quad \tau_a(C|s_i) = \tau_a(C|s_j), \quad \forall a \in \mathcal{A}, \forall C \in \mathcal{S}_B$$

where $\mathcal{S}_B$ is the state partition of equivalence classes defined by $B$. Two states $s_i, s_j \in \mathcal{S}$ are *bisimilar* if there exists a bisimulation relation $B$ such that $(s_i, s_j) \in B$. The largest $B$ is denoted as $\sim$.

The bisimulation relation is a rigid concept of state equivalence as it requires the exact equivalence of the reward and the transition probabilities for any pair of bisimilar states. Instead, the *bisimulation metric* (Ferns et al., 2004; 2011) measures this equivalence relation as an approximation and is defined as an operator $\mathcal{F} : \mathbb{M} \to \mathbb{M}$, where $\mathbb{M}$ is the set of all pseudometrics on $\mathcal{S}$, by:

$$\mathcal{F}(d)(s_i, s_j) = \max_{a \in \mathcal{A}} \left( \left| R(s_i, a) - R(s_j, a) \right| + \gamma W_1(d)(\tau_a(\cdot|s_i), \tau_a(\cdot|s_j)) \right) \tag{1}$$

where $d \in \mathbb{M}$ is a pseudometric, $W_1$ is the 1-Kantorovich (Wasserstein) metric measuring the distance between the transition probabilities. Ferns et al. (2004; 2011) show that $\mathcal{F}$ has a unique fixed point $d_\sim \in \mathbb{M}$ that is a bisimulation metric. $\mathcal{F}$ can be iteratively used to compute $d_\sim$, starting from an initial state $d_0$ and applying $d_{n+1} = \mathcal{F}(d_n) = \mathcal{F}^{n+1}(d_0)$. Ferns et al. (2011) also show that the bisimulation metric provides an upper bound on the difference between the optimal value functions:

$$|V^*(s_i) - V^*(s_j)| \le d_\sim(s_i, s_j) \tag{2}$$

Bisimulation relations require equivalence under all actions, even actions that may lead to negative outcomes, whereas we care about optimal actions. Castro (2020) defines the on-policy bisimulation relation, referred to as the $\pi$-bisimulation relation, that takes the behavioral policy into account when measuring behavioral similarity by considering the policy-induced dynamics and reward:

**Definition 2** ($\pi$-Bisimulation). A $\pi$-*bisimulation relation* on an MDP $\mathcal{M}$ is an equivalence relation $B^\pi \subseteq \mathcal{S} \times \mathcal{S}$ such that if $s_i B^\pi s_j$ holds for $s_i, s_j \in \mathcal{S}$, then the following properties are true:

$$R^\pi(s_i) = R^\pi(s_j) \quad \text{and} \quad \tau^\pi(C|s_i) = \tau^\pi(C|s_j), \quad \forall C \in \mathcal{S}_{B^\pi}$$

where $R^\pi(s) = \sum_{a \in \mathcal{A}} \pi(a|s) R(s, a)$, $\tau^\pi(C|s) = \sum_{a \in \mathcal{A}} \pi(a|s) \sum_{s' \in C} \tau_a(s'|s)$, and $\mathcal{S}_{B^\pi}$ is the state partition of equivalence classes defined by $B^\pi$.

Building on the work of Ferns et al. (2004; 2011), Castro (2020) defines the operator $\mathcal{F}^\pi$ as:

$$\mathcal{F}^\pi(d)\big(s_i, s_j\big) = \big|R^\pi(s_i) - R^\pi(s_j)\big| + \gamma W_1(d)\big(\tau^\pi(\cdot|s_i), \tau^\pi(\cdot|s_j)\big), \tag{3}$$

where $\mathcal{F}$ has a least fixed point $d_\sim^\pi$ that is also the $\pi$-*bisimulation metric*. Note that compared to Equation (1), the $\max_{a \in \mathcal{A}}$ operator is dropped because we are considering actions according to $\pi$. Moreover, Castro (2020) obtains the upper bound on the difference between the value functions as:

$$|V^\pi(s_i) - V^\pi(s_j)| \leq d_\sim^\pi(s_i, s_j) \tag{4}$$

## 3 PROBLEM FORMULATION

Fairness in ML entails ensuring unbiased decision making, and is generally categorized into individual and group fairness. While individual fairness aims to treat individuals similarly, group fairness focuses on ensuring that the distribution of outcomes is similar across different groups (Mehrabi et al., 2021). In this work, we specifically adopt group fairness, where a group is defined as:

**Definition 3** (Group). A *group* is a population associated with the sensitive attribute $g \in \mathcal{G}$.

In the above definition, a sensitive attribute can include factors such as race, gender, sexual orientation, etc. We further make the following assumptions regarding the sensitive attributes:

**Assumption 1.** Sensitive attributes $\mathcal{G}$ are observable to the decision making algorithm.

**Assumption 2.** Sensitive attributes $\mathcal{G}$ and group memberships remain constant during training.

These assumptions are commonly made in prior works on fairness in RL (Jabbari et al., 2017; Wen et al., 2021; Satija et al., 2023; Yin et al., 2023; Xu et al., 2024). Notably, prior works on fairness have showed that removing sensitive attributes from the decision making process, also known as "fairness through unawareness", is largely ineffective (Pedreshi et al., 2008; Barocas et al., 2023). Building upon the assumptions above, we define group-conditioned MDPs as:

**Definition 4** (Group-conditioned MDP). A *group-conditioned MDP* is a 6-tuple:

$$\mathcal{M}_{\text{group}} = (\mathcal{S}, \mathcal{A}, \mathcal{G}, \tau_a : \mathcal{S} \times \mathcal{A} \times \mathcal{G} \to \text{Dist}(\mathcal{S}), R : \mathcal{S} \times \mathcal{A} \times \mathcal{G} \to \mathbb{R}, \gamma)$$

where $\mathcal{S}$ is the state space, $\mathcal{A}$ is the action space, and $\mathcal{G}$ represents the sensitive attribute space. The group-specific transition dynamics are denoted by $\tau_a(s' \mid s, g)$, and $\text{Dist}(\mathcal{S})$ is the probability simplex over $\mathcal{S}$. The reward function specific to each group is $R(s, a, g)$, and $\gamma \in (0, 1]$ is the discount factor. The stationary policy is represented by $\pi(a|s, g)$, and the group-specific value function is defined as: $V^\pi(s, g) = \mathbb{E}_\pi \left[ \sum_{k=0}^\infty \gamma^k R(S_{t+k}, A_{t+k}, g) \mid S_t = s, G = g \right]$ for $s \in \mathcal{S}$ and $g \in \mathcal{G}$. The return of the policy is the expected return, given by: $J^\pi = \mathbb{E}_{s, g \sim \rho^\pi(s, g)}[V^\pi(s, g)]$ where $s, g$ are sampled from the specific stationary state-group distribution $\rho^\pi(s, g)$ according to $\pi$.

We use *demographic parity* (Dwork et al., 2012; Satija et al., 2023) as the group fairness definition. Informally, demographic parity requires that different groups should have similar returns. Formally, this fairness constraint is defined by Satija et al. (2023) as follows:

**Definition 5** (Demographic parity fairness in RL (Satija et al., 2023)). For some $\epsilon \geq 0$, denoting the acceptable margin of error, a policy $\pi$ satisfies demographic parity fairness at state $s$ if:

$$|J^\pi(s, g_i) - J^\pi(s, g_j)| \leq \epsilon, \quad \forall g_i, g_j \in \mathcal{G}.$$

The demographic parity notion aims to prevent disparate impact, where one group experiences significantly different outcomes than another. As an example, we can consider a credit scoring model that provides similar approval rates for different racial, gender, or socioeconomic groups. We refer to Satija et al. (2023) for a detailed discussion on the applicability and limitations of Definition 5.

## 4 BISIMULATION METRICS FOR LONG-TERM FAIRNESS IN RL

Our overarching goal is to develop a method that allows unconstrained policy optimization to inherently satisfy the fairness constraint. Rather than imposing the demographic parity constraint of Definition 5 or other fairness measures during policy optimization, we aim to adjust the reward

and observation dynamics of the MDP guided by the bisimulation metric. To achieve this, we first establish the connection between bisimulation metrics and the demographic parity fairness in RL.

Our objective is to make the group-conditioned MDP from Definition 4 behave as closely as possible for each group under a group-conditioned behavioral policy $\pi(a|s, g)$ over a long-term period. The $\pi$-bisimulation relation (Definition 2) is a natural fit for this goal as it essentially captures behavioral similarity induced by a given policy. To that end, we develop a conditional form of the $\pi$-bisimulation relation (Castro, 2020) that takes the sensitive attributes into account:

**Definition 6** (Group-conditioned $\pi$-Bisimulation). A *group-conditioned $\pi$-bisimulation relation* on an MDP $\mathcal{M}_{\text{group}}$ is an equivalence relation $B_{\text{group}}^\pi \subseteq \mathcal{S} \times \mathcal{G} \to \mathcal{S} \times \mathcal{G}$ such that if $(s_i, g_i)B_{\text{group}}^\pi(s_j, g_j)$ holds for $(s_i, g_i), (s_j, g_j) \in \mathcal{S} \times \mathcal{G}$, then the following properties are true:

$$R^\pi(s_i, g_i) = R^\pi(s_j, g_j) \quad \text{and} \quad \tau^\pi(C|s_i, g_i) = \tau^\pi(C|s_j, g_j), \quad \forall C \in \mathcal{S}_{B_{\text{group}}^\pi}$$

where $R^\pi(s, g) = \sum_{a \in \mathcal{A}} \pi(a|s, g)R(s, a, g)$, $\tau^\pi(C|s, g) = \sum_{a \in \mathcal{A}} \pi(a|s, g) \sum_{s' \in C} \tau_a(s'|s, g)$, and $\mathcal{S}_{B_{\text{group}}^\pi}$ is the partition of equivalence classes on the Cartesian product $\mathcal{S} \times \mathcal{G}$ defined by $B_{\text{group}}^\pi$.

Building on definitions of Castro (2020), we extend the operator $\mathcal{F}^\pi$ to a group-conditional variant:

$$\mathcal{F}_{\text{group}}^\pi(d)(s_i, g_i), (s_j, g_j) = |R^\pi(s_i, g_i) - R^\pi(s_j, g_j)| + \gamma W_1(d)(\tau^\pi(s_i'|s_i, g_i), \tau^\pi(s_j'|s_j, g_j)) \quad (5)$$

**Theorem 1.** $\mathcal{F}_{group}^\pi$ *as defined in Equation* (5) *has a least fixed point* $d_{group\sim}^\pi$, *and* $d_{group\sim}^\pi$ *is a group-conditioned $\pi$-bisimulation metric.*

The proof is in Appendix A.1 and consists of a reduction to the definitions of Castro (2020). The key idea allowing us to perform a reduction is that the sensitive attributes $g \in \mathcal{G}$ remain constant and have deterministic transitions. Similar to our work, the conditional form of $\pi$-bisimulation metrics has also been explored by Hansen-Estruch et al. (2022) in the context of goal-conditioned RL. Hansen-Estruch et al. (2022) used bisimulation for goal inference for robotic manipulation tasks. Here, we are defining the conditional form based on the sensitive attribute space which is not a subset of the state space, unlike the goal space in goal-conditioned RL.

**Theorem 2.** *For any two state-group pairs:*

$$|V^\pi(s_i, g_i) - V^\pi(s_j, g_j)| \leq d_{group\sim}^\pi((s_i, g_i), (s_j, g_j)) \quad (6)$$

The proof is in Appendix A.1 and follows the same logic as for Theorem 1. By comparing the result of Theorem 2 with the demographic fairness from Definition 5, we derive the following result:

**Theorem 3.** *Minimizing the bisimulation metric* $d_{group\sim}^\pi((s_i, g_i), (s_j, g_j))$ *results in demographic fairness as defined in Definition 5 between the two state-group pairs.*

The proof is in Appendix A.2 and is based on the convergence guarantees of the $\pi$-bisimulation metric. To achieve group fairness, we propose to reduce the group-conditioned $\pi$-bisimulation metric between state-group pairs for different groups in expectation over the stationary state distribution induced by the behavioral policy $\pi(a|s, g)$ by adjusting the reward function $J_{\text{rew.}}$ and observation dynamics $J_{\text{dyn.}}$. More formally, we propose to minimize:

$$J = \mathbb{E}_{\rho^\pi(s, g)} \Big[ \underbrace{|R^\pi(s_i, g_i) - R^\pi(s_j, g_j)|}_{J_{\text{rew.}}} + \underbrace{\gamma W_1(d_{\text{group}\sim}^\pi)(\tau^\pi(s_i'|s_i, g_i), \tau^\pi(s_j'|s_j, g_j))}_{J_{\text{dyn.}}} \Big] \quad (7)$$

where $\rho^\pi(s, g)$ is the stationary state-group distribution under the policy $\pi$. Notably, we use quantile matching to select state pairs from the group distributions. Quantile matching is a well-known statistical technique to map quantiles of two or more different populations for statistical analysis (McKay et al., 1979). In this context, we compare samples from corresponding quartiles of the population across different groups. This approach is essential because, in many cases, the state distributions of the groups may have little to no overlap. As we can split the expectation of Equation (7) into two terms $J = J_{\text{rew.}} + J_{\text{dyn.}}$, in subsequent sections, we outline practical algorithms for optimization of each term alongside the policy optimization.

### 4.1 BISIMULATION-DRIVEN OPTIMIZATION OF THE REWARD FUNCTION

We first describe our approach for optimization of the reward function by minimizing $J_{\text{rew.}}$:

$$J_{\text{rew.}} = \mathbb{E}_{s_i, s_j, g_i, g_j \sim \rho^\pi(s,g)} \left[ |R^\pi(s_i, g_i) - R^\pi(s_j, g_j)| \right] \tag{8}$$

This approach is closely related to bi-level optimization methods for reward shaping (Hu et al., 2020), however, the novelty of our method is that the reward shaping procedure is guided by the $\pi$-bisimulation metric. We assume the reward function $R(s, a, g)$ consists of the following terms:

$$R(s, a, g) = R^{\text{original}}(s, a) + \alpha R^{\text{correction}}_\phi(s, a, g) \tag{9}$$

where the first term is defined by the original MDP and is fixed; besides, this reward term is often not conditioned on the group membership. The second term is a learnable group-conditioned function, parameterized by $\phi$, that is used as a correction for the original reward, and $\alpha$ is a scalar weight.

Since modifying the reward function during the RL training may result in instability, our method learns the reward correction term outside the policy optimization loop. We take a sampling-based approach for minimizing $J_{\text{rew.}}$; first, we collect a dataset of trajectories using the policy $\pi$, then we use Equation (8) to estimate the discrepancy between the reward functions among different state-group pairs using quantile matching. Consequently, we optimize the estimated loss with respect to the learnable reward parameters $\phi$ using a gradient-based optimizer.

### 4.2 BISIMULATION-DRIVEN OPTIMIZATION OF THE OBSERVATION DYNAMICS

We now describe our approach for optimization of the observation dynamics by minimizing $J_{\text{dyn.}}$:

$$J_{\text{dyn.}} = \mathbb{E}_{s_i, s_j, g_i, g_j \sim \rho^\pi(s,g)} \left[ \gamma W_1(d^\pi_{\text{group}\sim})(\tau^\pi(s'_i | s_i, g_i), \tau^\pi(s'_j | s_j, g_j)) \right] \tag{10}$$

Critically, these modifications are carried out by the agent and only affect the observation space, leaving the underlying dynamics of the environment unchanged. In this approach, we assume that the observation dynamics has modifiable parameters $\omega$, examples of which are provided in Section 5. Notably, many real-world problems allow these types of modifications to the observations; for instance, a bank can consider to override the credit score of a loan applicant under certain circumstances (FDIC, 2005). Similarly to Section 4.1, we take a sampling-based approach for minimizing $J_{\text{dyn.}}$ while ensuring the stability of training. First, we collect a dataset of trajectories using the policy $\pi$, then we train a group-conditioned dynamics model $\mathcal{T}_\psi(s'|s, a, g)$ that outputs a normal distribution over the next state. For an efficient method of evaluating the Kantorovich metric in Equation (10), we follow Zhang et al. (2020) and substitute the distance measure with 2-Wasserstein ($W_2$) which has an analytical solution for normal distributions:

$$W_2\big(\mathcal{N}(\mu_1, \sigma_1), \mathcal{N}(\mu_2, \sigma_2)\big)^2 = \|\mu_1 - \mu_2\|_2^2 + \|\sigma_1^{\frac{1}{2}} - \sigma_2^{\frac{1}{2}}\|_F^2 \tag{11}$$

where $\mathcal{N}(\mu, \sigma)$ is a normal distribution, and $\|\cdot\|_F$ is the Frobenius norm. Since $J_{\text{dyn.}}$ is not differentiable with respect to the adjustable parameters $\omega$ in the MDP observation dynamics, we use gradient-free optimization methods to minimize this loss function. Note that unlike Section 4.1, we need to recollect the dataset of trajectories when the observation dynamics is modified.

### 4.3 BISIMULATOR: OPTIMIZATION OF THE REWARD FUNCTION AND OBSERVATION DYNAMICS

We can combine the algorithms outlined in Sections 4.1 and 4.2 to simultaneously optimize the reward function and observation dynamics of a given group-conditioned MDP so that its behaves $\pi$-bisimilarly for all groups, with the ultimate goal of achieving demographic fairness. The pseudo-code of our proposed method, referred to as the *Bisimulator*, is described in Algorithm 1. We can use any RL algorithm as the RL solver (L15), and we experiment with PPO (Schulman et al., 2017) and DQN (Mnih et al., 2015). We utilize Adam (Kingma & Ba, 2014) as the gradient-based optimizer of $J_{\text{rew.}}$ (L6), and use One-Plus-One (Juels & Wattenberg, 1995; Droste et al., 2002) as the gradient-free optimizer of $J_{\text{dyn.}}$ (L12). Additional implementation details are in Appendix D.

---

**Algorithm 1** Bisimulator: Optimization of the Reward Function and Observation Dynamics

---

**Inputs**: Reward optimization steps $M$, dynamics optimization steps $N$, learning steps $K$, and scalar weight $\alpha$.

1: Initialize policy $\pi_\theta(a|s, g)$, dynamics model $\mathcal{T}_\psi(s_i'|s_i, a_i, g_i)$, and reward function $R_\phi(s, a, g)$.
2: **while** not done **do**
3:     Collect dataset $\mathcal{D}$ of trajectories using $\pi_\theta(a|s, g)$ and the environment
4:     **for** optimization iteration $m = 1$ to $M$ **do**     $\triangleright$ Optimize the learnable reward function $R_\phi(s, a, g)$
5:         Estimate $J_{\text{rew}} \approx \mathbb{E}_\mathcal{D}\left[|R_{\text{orig.}}(s_i, a_i) + \alpha R_\phi(s_i, a_i, g_i) - R_{\text{orig.}}(s_j, a_j) - \alpha R_\phi(s_j, a_j, g_j)|\right]$
6:         $\phi \leftarrow \arg\min J_{\text{rew.}}$     $\triangleright$ Gradient-based optimization
7:     **end for**
8:     **for** optimization iteration $n = 1$ to $N$ **do**     $\triangleright$ Optimize parameters $\omega$ of the observation dynamics
9:         Collect dataset $\mathcal{D}$ of trajectories using $\pi_\theta$
10:       Train the dynamics model $\mathcal{T}_\psi(s'|s, a, g)$ using samples from $\mathcal{D}$
11:       Estimate $J_{\text{dyn}} \approx \mathbb{E}_\mathcal{D}\left[\gamma W_2(\mathcal{T}_\psi(s_i'|s_i, a_i, g_i), \mathcal{T}_\psi(s_j'|s_j, a_j, g_j))\right]$     $\triangleright$ Equation (11)
12:       $\omega \leftarrow \arg\min J_{\text{dyn.}}$     $\triangleright$ Gradient-free optimization
13:     **end for**
14:     **for** learning iteration $k = 1$ to $K$ **do**     $\triangleright$ Optimize the policy
15:       Update policy $\pi_\theta(a|s, g)$ using an RL algorithm
16:     **end for**
17: **end while**

---

## 5 EXPERIMENTAL RESULTS

Our experimental setup consists of sequential problems where fair decision making is crucial. We have utilized and extended a standard and well-established benchmark in this domain (D'Amour et al., 2020). Our aim is to showcase the versatility and applicability of our method, regardless of the specific fairness measures used, and importantly, without explicitly imposing those constraints.

As modifying the observation dynamics may not be feasible in certain real-world applications, we evaluate two variants of our method: the standard variant that optimizes both the reward and observation dynamics *(Bisimulator)*, and the variant that only optimizes the reward *(Bisimulator - Reward only)*. Furthermore, to showcase the versatility of our method across various RL algorithms, we apply Bisimulator to PPO (Schulman et al., 2017) and DQN (Mnih et al., 2015). All results are obtained on *10 seeds* and *5 evaluation episodes* per seed. Notably, we conducted grid search to tune the hyperparameters of all baselines, leading to an improvement over their original implementations.

### 5.1 CASE STUDY: LENDING

In this scenario, introduced by Liu et al. (2018), an agent representing the bank makes binary decisions on loan applications aimed at maximizing profit. The challenge is that these decisions result in changes in the population and their credit scores. Thus, even policies constrained to fairness measures at each time step can inadvertently increase the credit gap over a long-term horizon.

**Environment.** Each applicant has an observable group membership and a discrete credit score sampled from unequal group-specific initial distributions. At each time step, an applicant is sampled from the population, and the agent decides to accept or reject the loan. Successful repayment raises the applicant's credit score, benefiting the agent financially. Defaulting, however, reduces the credit score and the agent's profit. The probability of repayment in Liu et al. (2018); D'Amour et al. (2020) is a deterministic function of the applicant's credit score, however, this oversimplifies the actual dynamics of the problem[1] Therefore, we extend upon this by adding a latent variable representing the applicant's conscientiousness for repayment, regardless of their credit score. In both cases, an episode spans 10,000 steps and involves two groups, with the second group facing a disadvantage.

Finally, as an example of adjustable observation dynamics, described in Section 4.2, we utilize credit changes that depend on the applicant's group membership; for instance, applicants from the disadvantaged group may receive a higher credit increase upon loan repayment, compared to those who belong to the advantaged group. This is a realistic assumption since in practice, banks or other regulators are allowed to override credit scores during their decision making process (FDIC,

---

[1] A common counterexample is the population that is assigned a low credit score due to limited credit history, rather than their true likelihood of loan repayment.

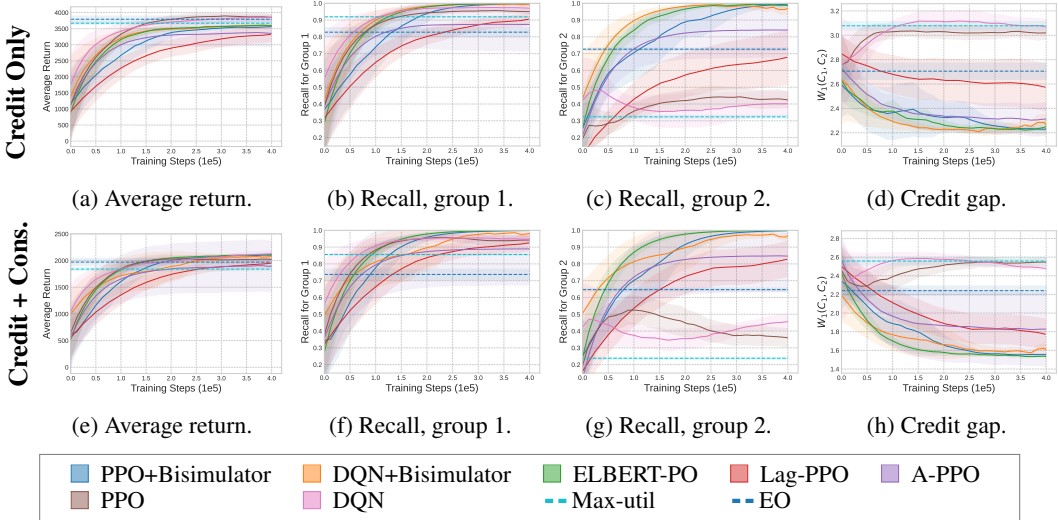

Figure 1: Lending results. The first row **(a-d)** shows the lending scenario where the repayment probability is only a function of the credit score, while the second row **(e-f)** presents the case where the repayment probability is a function of the credit score and a latent conscientiousness parameter. **(a, e)** Average return. **(b, f)** Recall for group 1. **(c, g)** Recall for group 2. **(d, h)** Credit gap measured as the Kantorovich distance between the credit score distributions at the end of evaluation episodes. The shaded regions show 95% confidence intervals and plots are smoothed for visual clarity.

2005). Importantly, these changes are on the agent side and only affect the observation dynamics, leaving the underlying dynamics and the probability of repayment unchanged. In other words, these modifications affect how the agent "sees" the world. Additional details are in Appendix B.1

**Fairness Metrics.** Similarly to D'Amour et al. (2020), we use three metrics for evaluating the long-term fairness: **(a)** changes in the credit score distributions measured by the Kantorovich distance, **(b)** the cumulative number of loans given to each group, and **(c)** agent's aggregated *recall*—$tp/(tp + fn)$—for loan decisions over the entire episode horizon, that is the ratio between the number of successful loans given to the number of applicants who would have repaid a loan.

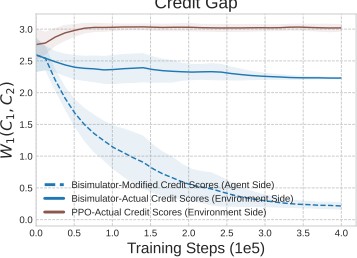

Credit Gap

Figure 2: Credit gaps of Bisimulator and PPO. Solid lines show the gap between the actual credit scores that govern the MDP dynamics, and the dashed line shows the gap between the modified credit scores that are observed by the agent.

**Baselines.** We evaluate our method against: a classifier that maximizes profits (Max-util) (Liu et al., 2018), an equality of opportunity (EO) classifier that maximizes profits constrained to equalized recalls (D'Amour et al., 2020), standard PPO and DQN, Lagrangian-PPO (Lag-PPO) (Satija et al., 2023) that is constrained to Definition 5, Advantage-regularized PPO (A-PPO) (Yu et al., 2022) that is constrained to equalized recalls, and ELBERT-PO (Xu et al., 2024), a recent state-of-the-art method that is constrained to equalized benefit rates. Additional details are in Appendix D.

**Results.** Figure 1 and Table 1 present the results of the two lending scenarios. Our method effectively achieves high recall values for both groups while narrowing down the credit gap. Notably, Bisimulator proves to be equally effective with both PPO and DQN, highlighting the versatility of our approach across different RL algorithms, unlike A-PPO or ELBERT-PO that are tightly coupled with PPO due to the modifications of the advantage function with fairness constraints. As anticipated, the greedy baselines (PPO, DQN, Max-util) obtain high recall values for group 1, but they fall short in achieving similar values for the disadvantaged group. A-PPO is constrained to small recall gaps, thus it naturally achieves low recall gaps, however, its recall values and credit gap are worse than those of Bisimulator. Bisimulator is able to match or surpass ELBERT-PO, the current

Table 1: Lending results. Reported values are the means and 95% confidence intervals, evaluated at the end of the training. Highlighted entries indicate the best values and any other values within 5% of the best value.

| | | Avg. Return | Credit Gap | Recall (G1) | Recall (G2) | Recall Gap |
|---|---|---|---|---|---|---|
| **Credit Only** | PPO + Bisimulator | 3582.63 ± 53.71 | **2.24 ± 0.05** | **1.00 ± 0.00** | **1.00 ± 0.00** | **0.00 ± 0.00** |
| | PPO + Bisimulator (Reward only) | 3568.20 ± 37.06 | **2.22 ± 0.04** | **1.00 ± 0.00** | **1.00 ± 0.00** | **0.00 ± 0.00** |
| | DQN + Bisimulator | 3547.02 ± 47.37 | **2.20 ± 0.05** | **0.99 ± 0.02** | **0.99 ± 0.02** | 0.01 ± 0.02 |
| | DQN + Bisimulator (Reward only) | 3590.27 ± 40.53 | **2.21 ± 0.04** | **0.99 ± 0.01** | **1.00 ± 0.00** | 0.01 ± 0.01 |
| | ELBERT-PO | 3636.42 ± 100.64 | **2.28 ± 0.10** | **1.00 ± 0.00** | **0.98 ± 0.03** | 0.02 ± 0.03 |
| | Lag-PPO | 3439.51 ± 237.18 | 2.52 ± 0.21 | 0.94 ± 0.03 | 0.72 ± 0.18 | 0.25 ± 0.16 |
| | A-PPO | 3365.82 ± 433.46 | **2.31 ± 0.15** | 0.87 ± 0.16 | 0.84 ± 0.17 | 0.06 ± 0.08 |
| | PPO | **3869.42 ± 113.24** | 3.02 ± 0.05 | **0.95 ± 0.01** | 0.42 ± 0.06 | 0.54 ± 0.06 |
| | DQN | **3849.40 ± 133.92** | 3.05 ± 0.06 | **0.97 ± 0.02** | 0.40 ± 0.07 | 0.56 ± 0.06 |
| | Max-util | 3670.66 ± 42.40 | 3.08 ± 0.04 | 0.92 ± 0.00 | 0.32 ± 0.01 | 0.60 ± 0.01 |
| | EO | **3793.72 ± 99.53** | 2.71 ± 0.07 | 0.83 ± 0.03 | 0.73 ± 0.01 | 0.10 ± 0.03 |
| **Credit + Cons.** | PPO + Bisimulator | **2116.16 ± 52.13** | **1.55 ± 0.04** | **1.00 ± 0.00** | **1.00 ± 0.00** | **0.00 ± 0.00** |
| | PPO + Bisimulator (Reward only) | **2082.24 ± 32.54** | **1.52 ± 0.05** | **1.00 ± 0.00** | **1.00 ± 0.00** | **0.00 ± 0.00** |
| | DQN + Bisimulator | **2085.93 ± 44.28** | **1.55 ± 0.03** | **0.99 ± 0.01** | **1.00 ± 0.00** | 0.01 ± 0.01 |
| | DQN + Bisimulator (Reward only) | **2128.07 ± 28.62** | **1.52 ± 0.04** | **0.99 ± 0.01** | **1.00 ± 0.00** | 0.01 ± 0.01 |
| | ELBERT-PO | **2110.56 ± 42.99** | **1.52 ± 0.03** | **1.00 ± 0.00** | **1.00 ± 0.00** | **0.00 ± 0.00** |
| | Lag-PPO | 2007.80 ± 90.66 | 1.70 ± 0.19 | **0.95 ± 0.05** | 0.87 ± 0.12 | 0.15 ± 0.11 |
| | A-PPO | 1915.77 ± 498.95 | 1.82 ± 0.42 | 0.89 ± 0.21 | 0.84 ± 0.22 | 0.05 ± 0.10 |
| | PPO | 2012.98 ± 70.02 | 2.54 ± 0.05 | 0.94 ± 0.02 | 0.35 ± 0.04 | 0.60 ± 0.07 |
| | DQN | **2131.00 ± 50.84** | 2.47 ± 0.04 | **0.95 ± 0.01** | 0.46 ± 0.05 | 0.49 ± 0.05 |
| | Max-util | 1840.06 ± 30.92 | 2.56 ± 0.04 | 0.86 ± 0.00 | 0.24 ± 0.01 | 0.62 ± 0.01 |
| | EO | 1971.54 ± 67.78 | 2.24 ± 0.05 | 0.74 ± 0.03 | 0.65 ± 0.01 | 0.09 ± 0.03 |

state-of-the-art method, demonstrating the effectiveness of our unconstrained approach in achieving long-term fairness. See Appendix C.1 for cumulative loans, the recall gap, and the results for Bisimulator (Reward only).

Generally, fairness interventions come at the expense of a decrease in the return, representing the bank's profit. Therefore, Bisimulator and fairness aware baselines expectedly achieve lower returns compared to the greedy ones. But interestingly, Bisimulator achieves similar or higher returns in the scenario with conscientiousness, showing its capability in handling more challenging scenarios.

To further shed light on how Bisimulator changes the observation dynamics, Figure 2 shows the credit gap between the groups for two sets of credit scores: the actual credit scores that govern the MDP dynamics and applicant's probability of repayment, and the agent-modified credit scores that only affect the observation space. The credit gap in the latter is much smaller, indicating that Bisimulator has indeed optimized the observation dynamics to favor fair outcomes. Interestingly, examining the optimized parameters reveals that Bisimulator has learned to provide higher credit increase upon loan repayment to the disadvantaged group and penalize them less upon loan default.

Finally, to demonstrate the scalability of our method to more complicated scenarios, Figure 3 and Table 2 present the results obtained for the lending scenario with 10 groups. In such problems, Equation (7) is evaluated and summed across all possible group pairs during a single update to optimize the reward and/or observation dynamics.

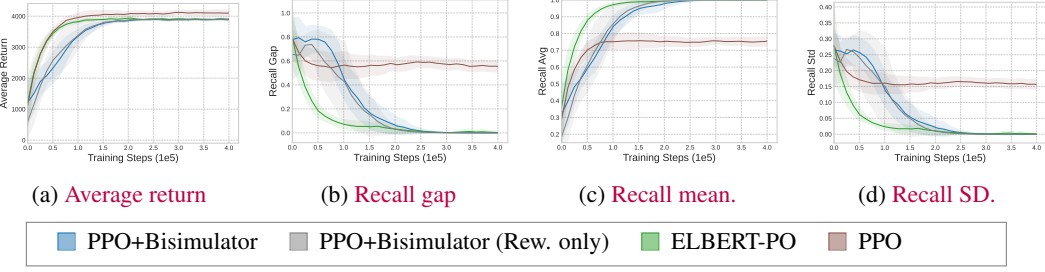

(a) Average return    (b) Recall gap    (c) Recall mean.    (d) Recall SD.

PPO+Bisimulator    PPO+Bisimulator (Rew. only)    ELBERT-PO    PPO

Figure 3: Lending results with 10 groups. **(a)** Average return. **(b)** Recall gap, **(c)** Mean, and **(d)** Standard deviation of the recall across all groups. The shaded regions show 95% confidence intervals and plots are smoothed for visual clarity.

Table 2: Lending results for 10 groups. Reported values are the means and 95% confidence intervals, evaluated at the end of the training. Highlighted entries indicate the best values and any other values within 5% of the best value.

|  | Avg. Return | Recall Mean | Recall SD | Recall Gap |
|---|---|---|---|---|
| PPO + Bisimulator | 3918.87 ± 67.58 | **1.00 ± 0.00** | 0.00 ± 0.00 | **0.00 ± 0.00** |
| PPO + Bisimulator (Reward only) | 3872.32 ± 86.35 | **1.00 ± 0.00** | 0.00 ± 0.00 | **0.00 ± 0.00** |
| ELBERT-PO | 3921.36 ± 62.30 | **1.00 ± 0.00** | 0.00 ± 0.00 | **0.00 ± 0.00** |
| PPO | **4127.90 ± 108.06** | 0.76 ± 0.03 | **0.15 ± 0.02** | 0.55 ± 0.07 |

## 5.2 CASE STUDY: COLLEGE ADMISSIONS

In this scenario, known as strategic classification (Hardt et al., 2016a), an agent representing the college makes binary decisions regarding admissions. The challenge arises when applicants can incur costs to alter their observable features, such as test scores. This manipulation disproportionately burdens individuals from disadvantaged groups who lack the financial means to afford these costs.

**Environment.** Each applicant has an observable group membership and a test score, along with an unobservable budget, both sampled from unequal group-specific distributions. At each time step, an applicant is sampled from the population and has a probability $\epsilon$ of being able to pay a cost to increase their score, provided their budget allows. The probability of success (e.g., the applicant eventually graduating) is a deterministic function of the true, unmodified score, and the agent's goal is to increase its accuracy in admitting applicants who will succeed. Importantly, since each applicant has a finite budget, over the episode horizon, the budget of the population decreases, hence making the problem sequential. Note that this environment is relatively different than that in (D'Amour et al., 2020) by having a more sequential nature due to its changing population. We study a scenario over 1,000 steps involving two groups, with group 2 facing a disadvantage.

As an example of adjustable observation dynamics, described in Section 4.2, we can consider group-specific costs for score modification. These adjustments can be seen as subsidized education for disadvantaged groups, a common practice. Additional details are in Appendix B.2.

**Fairness Metrics.** Following D'Amour et al. (2020), we use three metrics to assess fairness: **(a)** the *social burden* (Milli et al., 2019) that is the average cost individuals of each group have to pay to get admitted, **(b)** the cumulative number of admissions for each group, and **(c)** agent's aggregated *recall*—$tp/(tp + fn)$—for admissions over the entire episode horizon, that is the ratio between the number of admitted successful applicants to the number of applicants who would have succeeded.

**Baselines.** We evaluate our method against the same RL baselines described in Section 5.2. As a non-RL baseline, we employ a classifier that maximizes its accuracy through supervised learning, based on (D'Amour et al., 2020). Additional details are in Appendix D.

**Results.** Figure 4 and Table 3 show the results of the college admission environment. Bisimulator achieves the lowest recall gap and social burden for the disadvantaged group (group 2) compared to other methods. Similarly to Section 5.2, Bisimulator achieves equal performance when paired with

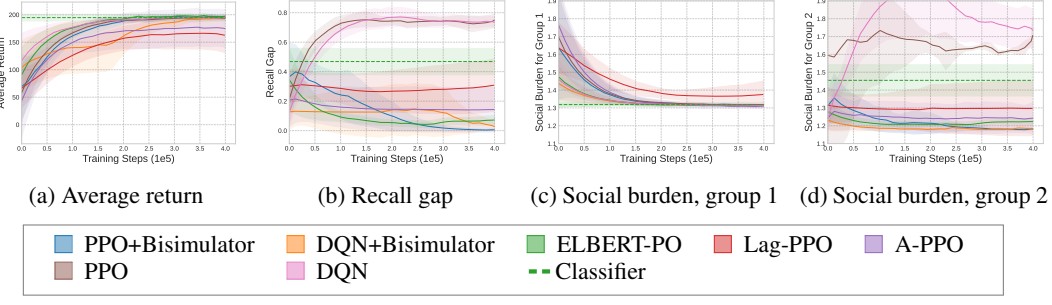

(a) Average return     (b) Recall gap     (c) Social burden, group 1     (d) Social burden, group 2

Figure 4: College admission results. The shaded regions show 95% confidence intervals and plots are smoothed for visual clarity.

Table 3: College admission results. Reported values are the means and 95% confidence intervals, evaluated at the end of the training. Highlighted entries indicate the best values and any other values within 5% of the best value. Social burden is abbreviated as Soc. Bdn.

| | Avg. Return | Soc. Bdn. (G1) | Soc. Bdn. (G2) | Recall (G1) | Recall (G2) | Recall Gap |
|---|---|---|---|---|---|---|
| PPO + Bisimulator | **192.42 ± 7.05** | **1.32 ± 0.01** | **1.18 ± 0.01** | **1.00 ± 0.00** | **1.00 ± 0.00** | **0.00 ± 0.00** |
| PPO + Bisimulator (Rew. only) | 191.34 ± 6.65 | **1.31 ± 0.01** | **1.16 ± 0.02** | **1.00 ± 0.00** | **1.00 ± 0.00** | **0.00 ± 0.00** |
| DQN + Bisimulator | **197.34 ± 6.93** | **1.32 ± 0.00** | **1.19 ± 0.01** | **1.00 ± 0.00** | **1.00 ± 0.00** | **0.00 ± 0.00** |
| DQN + Bisimulator (Rew. only) | **197.12 ± 9.91** | **1.31 ± 0.01** | **1.19 ± 0.01** | **1.00 ± 0.00** | 0.99 ± 0.02 | 0.01 ± 0.02 |
| ELBERT-PO | **201.60 ± 10.91** | **1.31 ± 0.00** | 1.22 ± 0.02 | **1.00 ± 0.00** | 0.92 ± 0.04 | 0.08 ± 0.04 |
| Lag-PPO | 151.94 ± 40.48 | 1.39 ± 0.10 | 1.30 ± 0.15 | 0.85 ± 0.17 | 0.79 ± 0.16 | 0.34 ± 0.20 |
| A-PPO | 172.30 ± 35.49 | **1.31 ± 0.01** | 1.25 ± 0.24 | 0.88 ± 0.17 | 0.74 ± 0.27 | 0.14 ± 0.15 |
| PPO | **193.92 ± 6.55** | **1.32 ± 0.01** | 1.83 ± 0.42 | **1.00 ± 0.00** | 0.22 ± 0.06 | 0.78 ± 0.06 |
| DQN | **197.04 ± 5.14** | **1.31 ± 0.01** | 1.69 ± 0.09 | **1.00 ± 0.00** | 0.28 ± 0.04 | 0.72 ± 0.04 |
| Classifier | **194.78 ± 6.13** | **1.32 ± 0.01** | 1.45 ± 0.09 | **1.00 ± 0.00** | 0.53 ± 0.09 | 0.47 ± 0.09 |

either DQN or PPO, demonstrating its applicability to various RL algorithms. See Appendix C.2 for cumulative admissions, recall values, and the results for Bisimulator (Reward only).

Analyzing the optimized parameters of the observation dynamics shows that Bisimulator has successfully learned to lower the cost of score modifications for the disadvantaged group. This aligns with the expected behavior, aiming to reduce the social burden on individuals of that group.

## 6 RELATED WORK

**Fairness in Sequential Decision Making.** In recent years, there has been a growing emphasis on the significance of dynamic analysis of fairness measures (Nashed et al., 2023). However, the exploration of these issues remains relatively restricted. The majority of existing studies focus on investigating fairness in multi-armed bandits (Liu et al., 2017; Joseph et al., 2016; Do et al., 2022; Metevier et al., 2019; Bistritz et al., 2020; Hossain et al., 2021). While the simplicity of the bandit problem allows for easier theoretical analysis, its practical applications often extend no further than recommender systems, failing to fully encompass the broader spectrum of real-world applications. In the context of RL, Jabbari et al. (2017) have proposed a fairness constraint suitable for the MDP setting, while providing a provably fair algorithm under an approximate notion of this constraint. Similarly, in the majority of the recently proposed approaches, fairness notions are adapted from the supervised learning setting and imposed as constraints during training of the optimal policy (Wen et al., 2021; Yu et al., 2022; Satija et al., 2023; Yin et al., 2023; Hu et al., 2023; Frauen et al., 2024). The recently proposed method of Xu et al. (2024) has adapted the concept of benefit rates to the RL setting and has demonstrated state-of-the-art performance. Another set of approaches use multi-objective MDPs (Siddique et al., 2020; Blandin & Kash, 2024), causal inference (Nabi et al., 2019), or the concept of welfare (Cousins et al., 2024; Yu et al., 2023). Finally, fairness is particularly important in multi-agent MDPs to ensure an optimal agent does not hinder the performance of other agents (Zhang & Shah, 2014; Jiang & Lu, 2019; Mandal & Gan, 2022; Ju et al., 2023).

**Optimization of MDP Reward (Reward Shaping).** Reward shaping is a technique involving the optimization of the reward signal to encourage desirable behaviors and discourage undesirable ones, ultimately leading to more effective learning (Ng et al., 1999). Common approaches include potential-based (Ng et al., 1999; Devlin & Kudenko, 2012; Gao & Toni, 2015), heuristics-based (Cheng et al., 2021), intrinsic motivation (Chentanez et al., 2004; Singh et al., 2010), bi-level optimization (Hu et al., 2020), and gradient-based (Sorg et al., 2010; Zheng et al., 2018) methods. Our proposed approach is closest to the bi-level optimization of Hu et al. (2020), however, the novelty of our approach is that the reward shaping procedure is guided by the bisimulation metric.

**Optimization of MDP Dynamics.** In contrast to the extensively explored concept of reward shaping, the optimization of MDP dynamics remains relatively less investigated. This disparity could be due to its stricter prerequisites, necessitating access to certain parameters within the dynamics model. The predominant focus in this domain revolves around the control and co-optimization of robots (Bächer et al., 2021; Spielberg et al., 2019; 2021; Ma et al., 2021; Wang et al., 2022; 2023; Evans et al., 2022). These works primarily aim to achieve an enhanced performance by concurrently learning to control a robot and optimizing its design and dynamical properties. Given the intertwined

nature of learning and optimization, the problem poses significant challenges, leading to the proposition of both gradient-based (Spielberg et al., 2019; Hu et al., 2019) and gradient-free (Cheney et al., 2018) optimization techniques. Notably, our method only optimizes the observation dynamics and leaves the underlying transitions, that affect the inherent behavior of the system, unchanged.

## 7 Broader Impact and Limitations

Addressing fairness in machine learning algorithms holds significant promise for promoting social justice and equity in various domains. By mitigating disparities, our proposed algorithm improves fairness in sequential decision making processes. However, it is important to acknowledge the limitations of our simulated experiments, which are based on simplified problems that may not fully capture real-world complexities. While we recognize the need for more sophisticated benchmarks, developing them is beyond the scope of this paper. Instead, we have utilized and extended the only well-established benchmark in this area (D'Amour et al., 2020), which has been widely used in recent studies (Xu et al., 2024; Deng et al., 2024; Hu et al., 2023; Yu et al., 2022).

Additionally, in this work, our focus is on group fairness, particularly the notion of demographic parity (Dwork et al., 2012) and its adaptation to RL (Satija et al., 2023). Our method's consistent success across various scenarios and metrics confirms that the demographic parity definition has broad applicability and effectiveness, laying a solid foundation for future research into other fairness notions. Finally, convergence proofs for RL methods based on $\pi$-bisimulation metrics are an open topic of research (Kemertas & Aumentado-Armstrong, 2021). It requires an intricate analysis on how the fixed-point properties of $\pi$-bisimulation interact with the convergence properties of a bisimulation-dependent policy, as they both rely on one another. This is an interesting research avenue on its own, beyond the primary focus of our paper, which is the application of bisimulation metrics for group fairness. Nonetheless, our approach and other methods (Zhang et al., 2020) have demonstrated strong and consistent empirical performance.

## 8 Conclusion

In this paper, we established the connection between bisimulation metrics and group fairness in reinforcement learning. Based on this insight, we proposed a novel approach that optimizes the reward function and observation dynamics of an MDP such that unconstrained optimization of the policy inherently results in the satisfaction of the fairness constraint. Crucially, these adjustments are carried out by the agent or a third-party regulator, without modifying the original MDP or its dynamics. A significant advantage of our approach is that it does not require modifying the underlying reinforcement learning algorithms, hence preserving the integrity of current decision making algorithms. Our method outperforms strong baselines on a standard fairness benchmark, highlighting its effectiveness.

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

## A  PROOFS

### A.1  BISIMULATION

**Bisimulation**  Bisimulation is a fundamental concept in concurrency theory (Larsen & Skou, 1991). It defines an equivalence relation between state-transition systems, ensuring that two systems can simulate each other's long-term behavior and remain indistinguishable to an external observer. Our work builds on the established theory of bisimulation developed by Larsen & Skou (1991); Desharnais et al. (2002); Ferns et al. (2004; 2011); Castro (2020), among others. Notably, we do not fully explore the potential of the conditional form of bisimulation metrics in this work. Our definitions, similarly to Hansen-Estruch et al. (2022), possess specific properties that allow us to reduce them to existing definitions. A comprehensive examination of the conditional form of bisimulation should be addressed as a standalone topic, as it lies beyond the scope of this work.

Given that our work extensively relies on the concept of metric spaces, we provide a summary of their definition below for completeness. For a more detailed introduction, we refer the reader to the existing literature and the work of Panangaden (2009).

A metric space is a pair $(X, d)$, where $X$ is a set and $d : X \times X \to \mathbb{R}_{\geq 0}$ is a function satisfying the following properties: (i) $\forall x, y \in X, d(x, y) = 0$ if and only if $x = y$ (identity), (ii) $\forall x, y \in X, d(x, y) = d(y, x)$ (symmetry), and (iii) $\forall x, y, z \in X, d(x, z) \leq d(x, y) + d(y, z)$ (triangle inequality). If $d$ satisfies these properties, it is called a *metric*; if the identity property is relaxed, it is called a *pseudometric*. The bisimulation metrics defined in this work are pseudometrics, as they relax the identity property—specifically, $d(s_i, s_j) = 0$ when $s_i$ and $s_j$ are behaviorally indistinguishable, but not necessarily when $s_i = s_j$. With this foundation, we can now proceed with the proofs of the definitions.

For convenience, we restate Theorem 2 of Castro (2020) using our notation.

Define $\mathcal{F}^\pi : \mathbb{M} \to \mathbb{M}$ by $\mathcal{F}^\pi(d)(s, t) = |R^\pi(s) - R^\pi(t)| + \gamma W_1(d)(\tau^\pi(s), \tau^\pi(t))$. Then, $\mathcal{F}^\pi$ has a least fixed point $d_\sim^\pi$, and $d_\sim^\pi$ is a $\pi$-bisimulation metric.

**Theorem 1.** $\mathcal{F}_{group}^\pi$ *as defined in Equation (5) has a least fixed point $d_{group\sim}^\pi$, and $d_{group\sim}^\pi$ is a group-conditioned $\pi$-bisimulation metric.*

*Proof.* Consider the MDP $\mathcal{M}_\mathcal{G} = (\mathcal{S}, \mathcal{A}, \mathcal{G}, \tau_a, R, \gamma)$. Define a new MDP $\overline{\mathcal{M}}_\mathcal{G} = (\overline{\mathcal{S}}, \mathcal{A}, \overline{\tau}_a, \overline{R}, \gamma)$, where $\overline{\mathcal{S}} = \mathcal{S} \times \mathcal{G}$, $\overline{\tau}_a : \overline{\mathcal{S}} \times \mathcal{A} \to \text{Dist}(\overline{\mathcal{S}})$, and $\overline{R} : \overline{\mathcal{S}} \times \mathcal{A} \to \mathbb{R}$. We can rewrite $\mathcal{F}_{\text{group}}^\pi$ from Equation (5) as follows:

$$\mathcal{F}_{\text{group}}^\pi(d)(\overline{s}_i, \overline{s}_j) = \left| \overline{R}^\pi(\overline{s}_i) - \overline{R}^\pi(\overline{s}_j) \right| + \gamma W_1(d)(\overline{\tau}^\pi(\overline{s}_i' \mid \overline{s}_i), \overline{\tau}^\pi(\overline{s}_j' \mid \overline{s}_j))$$

The state transition function $\overline{\tau}_a$ now outputs the group membership for the next state, which remains constant by assumption in Definition 3. Thus, the transition probability for this variable is deterministic, allowing us to concatenate $\mathcal{S}$ and $\mathcal{G}$ without altering the original definitions.

This formulation of $\mathcal{F}_{\text{group}}^\pi$ matches Castro's definition of $\mathcal{F}^\pi$, and the remainder of the proof follows the same steps as in Theorem 2 of Castro (2020). In summary, this proof mimics the argument of Ferns et al. (2011), with the added demonstration that $\mathcal{F}^\pi$ is continuous. □

Similarly, we restate Theorem 3 of Castro (2020).

Given any two states $s, t \in S$ in an MDP $\mathcal{M}$, $|V^\pi(s) - V^\pi(t)| \leq d_\sim^\pi(s, t)$.

**Theorem 2.** *For any two state-group pairs:*

$$|V^\pi(s_i, g_i) - V^\pi(s_j, g_j)| \leq d_{group\sim}^\pi((s_i, g_i), (s_j, g_j)) \tag{6}$$

*Proof.* Consider the MDP $\mathcal{M}_\mathcal{G} = (\mathcal{S}, \mathcal{A}, \mathcal{G}, \tau_a, R, \gamma)$ and define the new MDP

$\overline{\mathcal{M}}_\mathcal{G} = (\overline{\mathcal{S}}, \mathcal{A}, \overline{\tau}_a, \overline{R}, \gamma)$, where $\overline{\mathcal{S}} = \mathcal{S} \times \mathcal{G}$, $\overline{\tau}_a : \overline{\mathcal{S}} \times \mathcal{A} \to \text{Dist}(\overline{\mathcal{S}})$, and $\overline{R} : \overline{\mathcal{S}} \times \mathcal{A} \to \mathbb{R}$. We can rewrite Equation (6) as:

$$|V^\pi(\overline{s}_i) - V^\pi(\overline{s}_j)| \leq d_{\text{group}\sim}^\pi(\overline{s}_i, \overline{s}_j)$$

This bound on the value function difference matches Castro's definition, and the remainder of the proof follows Theorem 3 in Castro (2020), using induction on the standard value update. □

## A.2 Demographic Fairness with Bisimulation

**Extending Demographic Fairness to Infinite Horizon.**    Satija et al. (2023) defines the notion of demographic fairness using the expected cumulative reward in a finite-horizon setting on finite state and action spaces. Similarly to the case studies presented in our work, one can easily imagine the number of applicable scenarios where such assumptions hold true. An advantage of using bisimulation metrics in this setting is that they are defined for infinite horizon. As such, we must extend the definition of Satija et al. (2023) to an infinite horizon case. To do so, we simply use the discounted expected cumulative return instead. More precisely, we use the definition of $J^\pi$ that includes a discount factor $\gamma \in (0, 1]$.

Given an MDP $\mathcal{M}_{group}$ as introduced in Definition 4, at a specific time step $t$, the return of the policy $J^\pi$ is as follows:

$$J^\pi = \sum_{s,g} \rho(s_t, g_t) \mathbb{E}_\pi \left[ \sum_{k=0}^{\infty} \gamma^k R(S_{t+k}, A_{t+k}, g) \mid S_t = s, G = g \right]$$

As opposed to Satija et al. (2023), who defines it for a horizon $H$ as:

$$J^\pi = \sum_{s,g} \rho(s_t, g_t) \mathbb{E}_\pi \left[ \sum_{k=0}^{H} R(S_{t+k}, A_{t+k}, g) \mid S_t = s, G = g \right]$$

**Bounding group-conditioned $\pi$-bisimulation metric.**    An important result that Castro (2020) shows in his work is the convergence of the $\pi$-bisimulation metric. Specifically, by assuming that we can sample transitions infinitely often, for a time step $t$, updating $\lim_{t\to\infty} d_t^\pi = d_\sim^\pi$ almost certainly. We use this result to bound $d_\sim^\pi$ by an arbitrary $\epsilon \in \mathbb{R}$.

**Achieving Demographic Parity Fairness.**    Given the previous statements, we can now derive the proof for Theorem 3.

**Theorem 3.** *Minimizing the bisimulation metric $d_{group\sim}^\pi((s_i, g_i), (s_j, g_j))$ results in demographic fairness as defined in Definition 5 between the two state-group pairs.*

*Proof.* We begin from the definition of demographic fairness as in Definition 5:

$$\begin{aligned}
|J^\pi(s_i, g_i) - J^\pi(s_j, g_j)| &= |\mathbb{E}_{\rho(s,g)}[V^\pi(s_i, g_i)] - \mathbb{E}_{\rho(s,g)}[V^\pi(s_j, g_j)]| \\
&\leq \mathbb{E}_{\rho(s,g)}\left[|V^\pi(s_i, g_i) - V^\pi(s_j, g_j)|\right] \\
&\leq \mathbb{E}_{\rho(s,g)}\left[d_{group\sim}^\pi\left((s_i, g_i), (s_j, g_j)\right)\right] \\
&\leq \epsilon
\end{aligned}$$

Where the second line follows from the triangle inequality. We can see that the third line follows from Theorem 2 and is exactly equal to our definition of $J$ in Equation (7). Then, since we can bind the group-conditioned $\pi$-bisimulation metric by an epsilon, it follows that minimizing the metric in expectation leads to minimizing the fairness bound. Thus, we can achieve fairness up to an acceptable margin of error $\epsilon$ using bisimulation metrics. □

## B    Environment Details

The code for the environments is included in the supplemental material, and will be made publicly available. These environments are accurate re-implementations of ml-fairness-gym (D'Amour et al., 2020). In comparison, our environments have additional features and more user-friendly implementations, and follow the updated Gymnasium (Towers et al., 2023) API rather than the deprecated OpenAI Gym (Brockman et al., 2016) interface.

### B.1    Lending Environment

**Environment.**    In the lending scenario, an agent representing the bank makes binary decisions loan applications with the goal of increasing its profit. Each applicant has an observable group membership $g \in \mathcal{G}$ and a discrete credit score $1 \leq c \leq C_{max}$ sampled from group-specific and unequal initial distributions $p_0(c|g)$. At each time step $t$, applicants are sampled uniformly with replacement from the population, and the agent decides to accept or reject the loan application. Successful repayment raises the applicant's credit score by $c_+$, benefiting the agent financially with $r_+$. Defaulting, however, reduces the credit score by $c_-$ and the agent's profit by $r_-$. If the agent rejects the loan, it receives no reward. As discussed in Section Section 5.1, we examine two variants of the lending scenario:

1. **Credit only:** The probability of repayment is a deterministic function of the applicant's credit score, similarly to D'Amour et al. (2020). However, this model oversimplifies certain dynamics, as the probability of repayment in reality can be a function of many factors beyond the credit score. Additionally, this model fails to capture the case where an individual is assigned a low credit score due to their limited credit history, rather than their true likelihood of loan repayment.
2. **Credit + Conscientiousness:** The probability of repayment is a function of the applicants credit score and an unobservable latent variable representing the applicants conscientiousness. The conscientiousness for each individual is sampled from a Normal distribution and is independent from their group membership.

The observation space in both variants include the applicant's credit score, group membership, the ratio of the past loan repayments, and the ratio of the past loan defaults. As discussed Section 4.2, the Bisimulator algorithm, is allowed to change the observation dynamics. In this scenario, Bisimulator changes the group-specific values for $c_+$ and $c_-$. For instance, applicants from the disadvantaged group may receive a higher credit increase upon loan repayment, compared to those who belong to the advantaged group. This is a realistic assumption since in practice, banks or other regulators are allowed to override credit scores during their decision making process (FDIC, 2005). Importantly, these changes are carried out by the agent and only affect the observation space, leaving the underlying dynamics and the probability of repayment unchanged. In other words, the changes are on the agent side and affect how it "sees" the observations and they do not impact the actual dynamics.

Figure 5 shows the initial credit score distribution for each group, and Table 4 presents additional details of this environment.

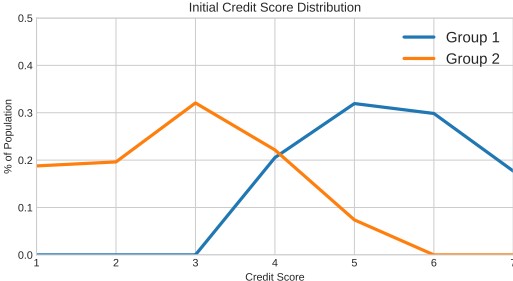

Figure 5: Initial credit score distribution for each group.

Table 4: Details of the lending environment.

| Parameter | Value |
|---|---|
| Number of groups | 2 |
| Group distributions | $(0.5, 0.5)$ |
| $C_{max}$ | 7 |
| $c_+$ and $c_-$ | $+1$ and $-1$ |
| $r_+$ and $r_-$ | $+1$ and $-1$ |
| Probability of repayment for each credit score | $(0.3, 0.4, 0.5, 0.6, 0.7, 0.8, 0.9)$ |
| Conscientiousness distribution | $\mathcal{N}(0.55, 0.1)$ |
| Population size | 1000 |
| Episode horizon (steps) | 10000 |

**Fairness Metrics** Following D'Amour et al. (2020), we use three metrics to assess fairness: **(a)** the *social burden* (Milli et al., 2019) that is the average cost individuals of each group have to pay to get admitted, **(b)** the cumulative number of admissions for each group, and **(c)** agent's aggregated *recall*—$tp/(tp + fn)$—for admissions over the entire episode horizon, that is the ratio between the number of admitted successful applicants to the number of applicants who would have succeeded.

## B.2 COLLEGE ADMISSIONS ENVIRONMENT

**Environment** In the college admissions scenario, an agent representing the college makes binary decisions regarding admissions. Each applicant has an observable group membership $g \in \mathcal{G}$ and a discrete test score $1 \leq c \leq C_{max}$, along with an unobservable budget $0 \leq b \leq B_{max}$, both sampled from unequal group-specific distributions $p_0(c|g)$ and $p_0(b|g)$. At each time step $t$, an applicant is sampled from the population and has a probability $\epsilon$ of being able to pay a cost to increase their score, provided their budget allows. The probability of success (e.g., the applicant eventually graduating) is a deterministic function of the true, unmodified score, and the agent's goal is to increase its accuracy in admitting applicants who will succeed. If the agent correctly admits an applicant, it receives a reward $r_+$ and if it rejects an applicant who would have been successful, it receives a reward of $r_-$, otherwise its reward is zero. If an applicant is admitted during an episode, it is no longer sampled. Importantly, since each applicant has a finite budget, over the episode horizon, the budget of the population decreases, hence making the problem sequential. Note that this environment is substantially different than that in (D'Amour et al., 2020) by having a more sequential nature due to its changing population.

As discussed Section 4.2, the Bisimulator algorithm, is allowed to change the observation dynamics. In this scenario, Bisimulator changes the group-specific costs for score modification. These adjustments can be seen as subsidized education for disadvantaged groups, a common practice. Importantly, these changes are carried out by the agent and only affect the observation space, leaving

Table 5: Details of the college admissions environment.

| Parameter | Value |
|---|---|
| Number of groups | 2 |
| Group distributions | $(0.5, 0.5)$ |
| $C_{max}$ | 10 |
| $B_{max}$ | 5 |
| $r_+$ and $r_-$ | $+1$ and $-1$ |
| Probability of success for each score | $(0.0, 0.1, 0.2, 0.3, 0.4, 0.5, 0.6, 0.7, 0.8, 0.9)$ |
| Probability of score modification ($\epsilon$) | 0.5 |
| Score distributions | Group 1: $\mathcal{N}(8, 1)$, Group 2: $\mathcal{N}(5, 1)$ |
| Budget distributions | Group 1: $\mathcal{N}(4, 1)$, Group 2: $\mathcal{N}(2, 1)$ |
| Population size | 1000 |
| Episode horizon (steps) | 1000 |

the underlying dynamics and the probability of success unchanged, since the probability of success is a function of the true, unchanged score. Table 5 presents additional details of this environment.

**Fairness Metrics**    Following D'Amour et al. (2020), we use three metrics to assess fairness: **(a)** the *social burden* (Milli et al., 2019) that is the average cost individuals of each group have to pay to get admitted, **(b)** the cumulative number of admissions for each group, and **(c)** agent's aggregated *recall*—$tp/(tp + fn)$—for admissions over the entire episode horizon, that is the ratio between the number of admitted successful applicants to the number of applicants who would have succeeded.

## C  ADDITIONAL EXPERIMENTAL RESULTS

This section includes additional experimental results to complement that of Section 5.

### C.1  CASE STUDY: LENDING

Figure 6 shows the cumulative loans given to each group over the course of evaluation episodes. While all methods regularly approve loans of the first group, Bisimulator and ELBERT-PO are giving an equal amount of loans to the second group while keeping high recall values (refer to Figure 1 and Table 1).

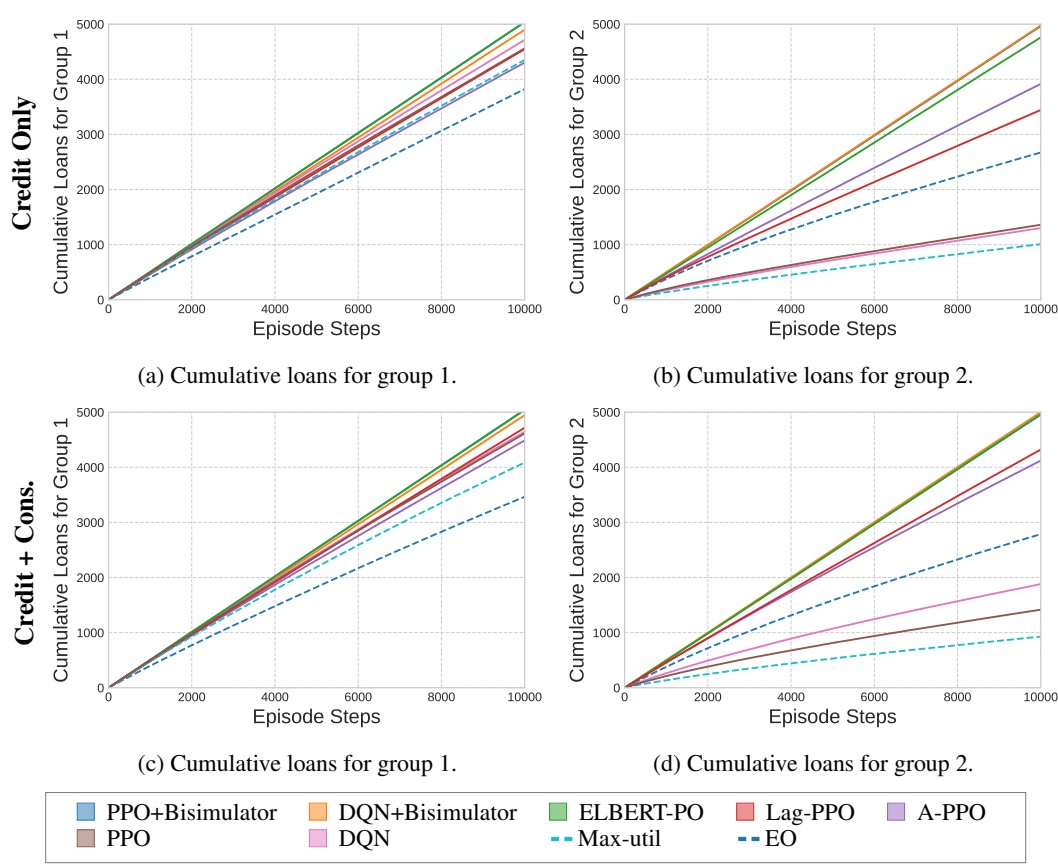

(a) Cumulative loans for group 1.

(b) Cumulative loans for group 2.

(c) Cumulative loans for group 1.

(d) Cumulative loans for group 2.

Figure 6: Lending results. Cumulative loans given to each group over the course of evaluation episodes. The first row **(a, b)** shows the lending scenario where the repayment probability is only a function of the credit score, while the second row **(c, d)** presents the case where the repayment probability is a function of the credit score and a latent conscientiousness parameter. Results are obtained on 10 seeds and 5 evaluations episodes per seed. Confidence intervals are not shown for visual clarity.

Figure 7 shows the recall gap between the two groups over the training steps. Since A-PPO and EO are explicitly constrained to minimize the recall gap, they achieve low recall gaps, similarly to Bisimulator. However, the recall values for each group are considerably lower than those of Bisimulator (refer to Figure 1 and Table 1).

Figure 8 presents a comparison between Bisimulator and Bisimulator (Reward Only), complementing the results in Table 1. Although optimizing both dynamics and rewards improves the overall performance, the variant focusing solely on reward optimization remains competitive.

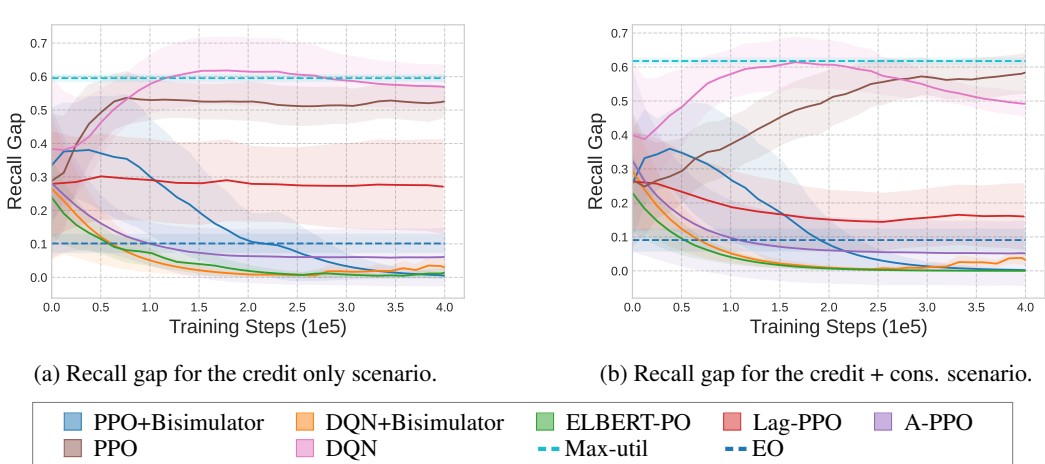

(a) Recall gap for the credit only scenario. (b) Recall gap for the credit + cons. scenario.

Figure 7: Lending results. Recall gaps between the two groups over the training steps. **(a)** shows the lending scenario where the repayment probability is only a function of the credit score, while the second row **(b)** presents the case where the repayment probability is a function of the credit score and a latent conscientiousness parameter. Results are obtained on 10 seeds and 5 evaluations episodes per seed. The shaded regions show 95% confidence intervals and plots are smoothed for visual clarity.

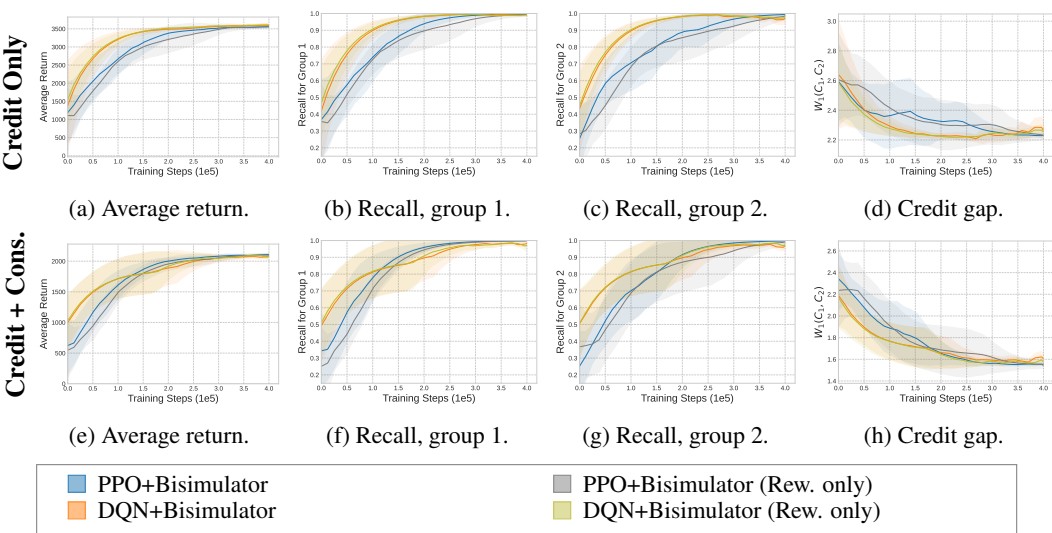

(a) Average return. (b) Recall, group 1. (c) Recall, group 2. (d) Credit gap.

(e) Average return. (f) Recall, group 1. (g) Recall, group 2. (h) Credit gap.

Figure 8: Comparison of Bisimulator and Bisimulator (Reward only). The first row **(a-d)** shows the lending scenario where the repayment probability is only a function of the credit score, while the second row **(e-f)** presents the case where the repayment probability is a function of the credit score and a latent conscientiousness parameter. **(a, e)** Average return. **(b, f)** Recall for group 1. **(c, g)** Recall for group 2. **(d, h)** Credit gap measured as the Kantorovich distance between the credit score distributions at the end of evaluation episodes. Results are obtained on 10 seeds and 5 evaluations episodes per seed. The shaded regions show 95% confidence intervals and plots are smoothed for visual clarity.

## C.2 CASE STUDY: COLLEGE ADMISSIONS

Figure 9 shows the cumulative admissions granted to each group over the course of evaluation episodes. All methods regularly accept applicants from group 1, however, only Bisimulator and ELBERT-PO are granting an equal amount of admissions to group 2 while keeping high recall values (refer to Figure 4 and Table 3).

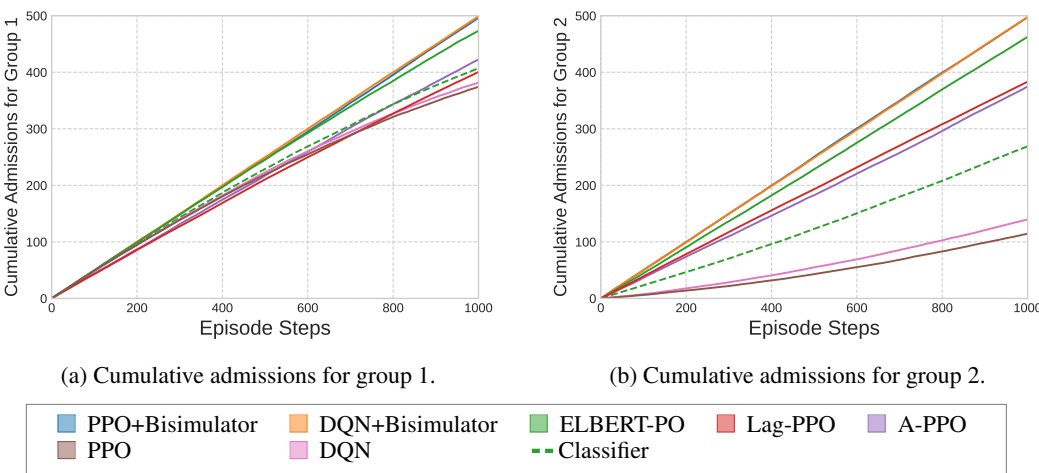

(a) Cumulative admissions for group 1.      (b) Cumulative admissions for group 2.

Figure 9: College admission results. Cumulative admissions granted to each group over the course of evaluation episodes. Results are obtained on 10 seeds and 5 evaluations episodes per seed. Confidence intervals are not shown for visual clarity.

Figure 10 shows the recall values for each group. Bisimulator obtains high recall values for both groups. Notably, the recall gap obtained by Bisimulator is the smallest among all the methods (refer to Figure 4 and Table 3).

Figure 11 presents a comparison between Bisimulator and Bisimulator (Reward Only), complementing the results in Table 3. Similarly to the lending experiments, optimizing both dynamics and rewards improves the overall performance, specifically in terms of recall gap. However, the variant focusing only on reward optimization remains competitive.

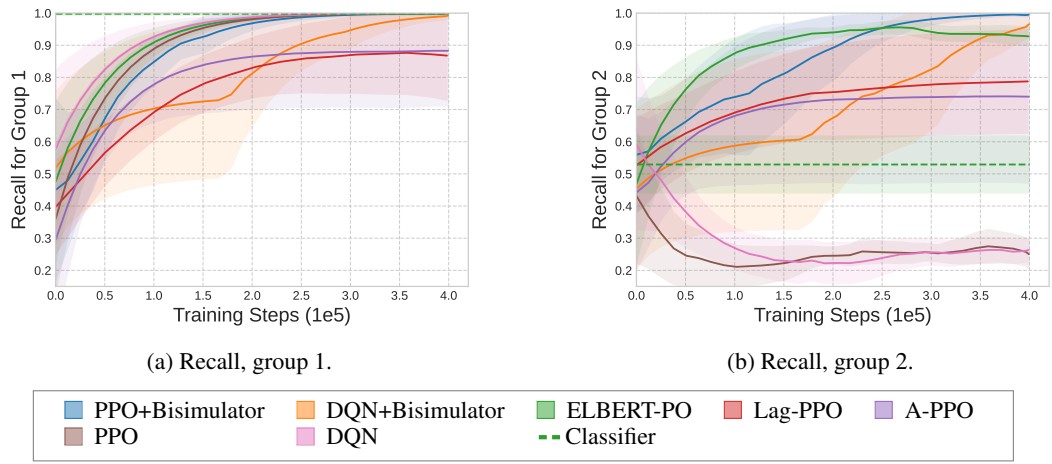

(a) Recall, group 1.      (b) Recall, group 2.

Figure 10: College admission results. Recall values for each group over the training steps. **(a)** Recall for group 1. **(b)** Recall for group 2. Results are obtained on 10 seeds and 5 evaluations episodes per seed. The shaded regions show 95% confidence intervals and plots are smoothed for visual clarity.

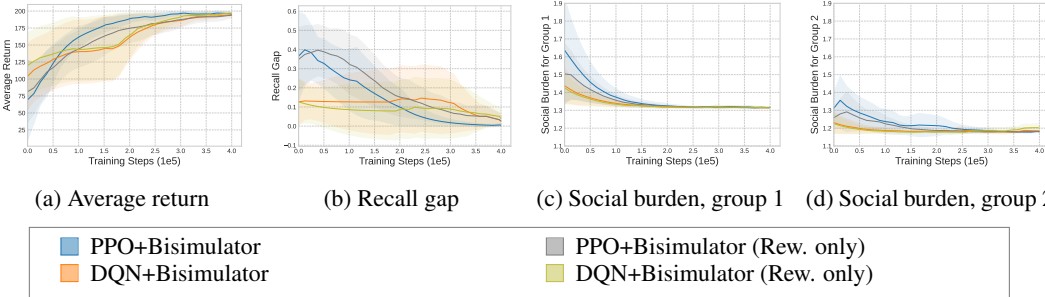

Figure 11: College admission results. **(a)** Average return. **(b)** Recall gap. **(c)** Social burden for group 1. **(d)** Social burden for group 2. Results are obtained on 10 seeds and 5 evaluations episodes per seed. The shaded regions show 95% confidence intervals and plots are smoothed for visual clarity.

## D    IMPLEMENTATION DETAILS

The codes for Bisimulator and all of the baselines is included in the supplemental material, and will be made publicly available.

### D.1    HYPERPARAMETERS

Our PPO and DQN implementations are based on CleanRL (Huang et al., 2022). We have further tuned their hyperparameters, listed in Tables 6 and 7, with grid search. The actor and critic have MLP networks with the Tanh activation function and one hidden layer with dimension of 256. As discussed in Section 4, one of the advantages of Bisimulator is that is has very few hyperparameters; Table 8 present these values. We use PPO and DQN as the RL backbone, utilize Adam (Kingma & Ba, 2014) as the gradient-based optimizer of $J_{\text{rew.}}$, and use One-Plus-One (Juels & Wattenberg, 1995; Droste et al., 2002) as the gradient-free optimizer of $J_{\text{dyn.}}$.

The dynamics model $\mathcal{T}_\psi(s'|s, a, g)$ in Algorithm 1 is implemented as an MLP that outputs a Gaussian distribution over the next state. Since the state space is discrete, we use straight-through-estimator (Bengio et al., 2013) to propagate the gradients.

Finally, as discussed in Section 4, we use quantile matching (McKay et al., 1979) to select the state-group pairs from the on-policy distribution. In practice, we use quartiles obtained on the batch of

Table 6: Hyperparameters for PPO.

| Hyperparameter | Setting |
|---|---|
| Optimizer | Adam |
| Hidden layer width | 256 |
| Learning rate | 5e-5 |
| Discount factor $\gamma$ | 0.99 |
| $\lambda$ for GAE | 0.95 |
| Batch size | 512 |
| Mini batch size | 64 |
| Policy update epochs | 5 |
| Surrogate clipping coefficient | 0.2 |
| Entropy coefficient | 0.01 |
| Value function coefficient | 0.5 |
| Maximum norm for gradient clipping | 0.5 |
| Clip value function loss | True |
| Anneal learning rate | True |

Table 7: Hyperparameters for DQN.

| Hyperparameter | Setting |
|---|---|
| Optimizer | Adam |
| Hidden layer width | 256 |
| Learning rate | 5e-5 |
| Discount factor $\gamma$ | 0.99 |
| Batch size | 512 |
| Target network update rate $\tau$ | 1 |
| Target network update frequency | 10 |
| Update epochs | 4 |
| Anneal learning rate | True |

the data. For example, the first quartile of group 1 is matched with the first quartile of group 2 in order to estimate $J_{\text{rew.}}$ and $J_{\text{dyn.}}$.

Table 8: Hyperparameters for Bisimulator in lending and college admission environments, to accompany Algorithm 1.

| Hyperparameter | Setting | |
|---|---|---|
| | PPO | DQN |
| Reward optimization iterations ($M$) | 1 | 1 |
| Observation dynamics optimization iterations ($N$) | 300 | 300 |
| Policy update iterations ($K$) | 1 | 1 |
| Reward coefficient ($\alpha$) | 5 | 1.5 |

## D.2 BASELINES

All of the baselines follow their official implementations. We started from the the suggested hyperparameters for each baseline and further tuned it with grid search for each environment. For a fair comparison among the deep RL algorithms that are based on PPO (Bisimulator+PPO, A-PPO, Lag-PPO, and ELBERT-PO), the architecture of the MLP networks and the hyperparameters of the PPO algorithm follow the details outlined in Table 6.

## D.3 COMPUTING INFRASTRUCTURE

Our results are obtained using Python v3.11.5, PyTorch v2.2.1 and CUDA 12.2. Experiments have been conducted on a cloud computing service with Nvidia V100 GPUs, Intel Gold 6148 Skylake CPU, and 32 GB of RAM. In this setting, each experiment takes between 1 to 2 hours for 400 thousands steps of training.

