# OpenReview forum: "Long-Term Fairness in Reinforcement Learning with Bisimulation Metrics"
_ICLR.cc/2025/Conference — Submitted to ICLR 2025_

### Official Review · Reviewer_c9Ni · 2024-10-29

**Soundness:** 3
**Presentation:** 3
**Contribution:** 2
**Rating:** 6
**Confidence:** 4

**Summary:**

This paper introduces a novel method for learning group-fair solutions in reinforcement learning (RL), addressing a limitation of standard deep RL approaches, which prioritize reward maximization over fairness considerations. The proposed method learns group-specific reward models and observation dynamics alongside the RL policy, using bisimulation metrics without modifying the core RL optimization process. Experiments in lending and college admission scenarios are presented that show the proposed method is comparable with considered baselines.

**Strengths:**

The paper presents a clear and well-structured approach to incorporating group-level fairness in RL.

The connection drawn between group fairness and bisimulation is both novel and thorough.

The paper introduces a practical implementation of their framework, incorporating algorithmic innovations that enable efficient computation.

The experimental results effectively demonstrate the method's ability to learn group fair solutions in two domains.

**Weaknesses:**

Despite the paper's compelling core idea, the empirical evaluation appears somewhat limited in scope, with only two relatively simple domains and discrete action spaces. I believe, incorporating more complex scenarios with multiple groups with conflicting nature would greatly improve the paper.

A detailed and thorough theoretical analysis is also lacking, particularly in terms of rigorously examining the properties of the proposed reward models and observation dynamics and their relationship to bisimulation metrics.

The paper's organization could also be improved, with a formal definition of long-term fairness and clearer connections between bisimulation and fairness metrics.

**Questions:**

1. Could you provide a clear definition of long-term fairness in the context of your work? This definition is missing and should have been defined formally.
2. How well would the proposed method perform in environments with more groups and actions? What are the theoretical and practical limitations in scaling the approach, and how might these be addressed?
3. Current results show DQN performing comparably or better than PPO, which contradicts common findings in the field. This pattern appears in both standard implementations and when combined with bisimulation. Could you explain this unexpected behavior, and is it possibly related to hyperparameter tuning?
4. How does your approach compare to welfare-based RL methods[1-5]? Specifically, wouldn't egalitarian or lexicographic egalitarian welfare functions, which maximize the minimum group utility, achieve similar goals in reducing recall and credit gaps?
5. Regarding the implementation details, how do you handle the optimization ordering between reward and observation dynamics models (assuming they're represented by neural networks)? Does the optimization order impact performance? Could you specify which gradient-free optimization method was employed?

[1] Cousins, Cyrus, Kavosh Asadi, and Michael L. Littman. "Fair E3: Efficient welfare-centric fair reinforcement learning." 5th Multidisciplinary Conference on Reinforcement Learning and Decision Making. RLDM. 2022.

[2]Siddique, Umer, Paul Weng, and Matthieu Zimmer. "Learning fair policies in multi-objective (deep) reinforcement learning with average and discounted rewards." International Conference on Machine Learning. PMLR, 2020.

[3] Zimmer, Matthieu, et al. "Learning fair policies in decentralized cooperative multi-agent reinforcement learning." International Conference on Machine Learning. PMLR, 2021.

[4] Fan, Zimeng, et al. "Welfare and fairness in multi-objective reinforcement learning." arXiv preprint arXiv:2212.01382 (2022).

[5] Cousins, Cyrus, et al. "On Welfare-Centric Fair Reinforcement Learning."

---

> ### Author Response · Authors · 2024-11-19
>
> Thank you for your review.
>
> ## Weaknesses
> * **[Simple environments]** As highlighted in the limitations section (Section 7), the benchmark we utilized in this paper (D'Amour et al., 2020) is currently the **only** established benchmark within the fair RL domain. Several recent works (Xu et al., 2024; Deng et al., 2024; Hu et al., 2023; Yu et al., 2022) have also relied on this same benchmark. While we acknowledge the need for more advanced benchmarks, developing such benchmarks falls outside the scope of our paper. If the reviewer is aware of a more complex benchmark that could enhance our evaluation, we would welcome the suggestion.
>
> * **[Theoretical analysis on the properties of the reward and observation dynamics]** We agree that a theoretical analysis in this area would be valuable. However, we would like to remind the reviewer that our focus is within the deep RL setting. Although our method is solidly based on bisimulation metrics, conducting a theoretical analysis of the learned reward function or transition dynamics is typically only feasible in finite MDPs or bandit settings—problems that are not the focus of our study. Instead, our primary objective is to present a **practical** algorithm with some theoretical foundation. That said, we have conducted empirical evaluations of the transition dynamics in Section 5 (Lines 374 and 463 of the initial version).
>
> * **[Paper’s organization]** We have revised the manuscript for improved clarity and have added the modified definition of long-term fairness in the introduction based on Yin et al. (2023). The reviewer mentioned a need for a clearer connection between bisimulation and fairness metrics, but it is unclear to us what specific aspect needs further clarification. We have explicitly established this connection through Theorems 1, 2, and 3 in Section 4, with detailed proofs provided in Appendix A. If there are specific questions or concerns regarding this connection, we would be happy to address them.

---

> > ### Author Response · Authors · 2024-11-19
> >
> > ## Questions
> > 1. **[Definition of long-term fairness]** We provided a brief definition of long-term fairness in Lines 30-31 of the original submission: "ensuring fairness over an extended period rather than at the level of individual decisions." A more formal definition has been added to the revised version, following Yin et al. (2023):
> > > Yin et al. (2023) define long-term fairness in RL as the optimization of the cumulative reward subject to a constraint on the cumulative utility, reflecting fairness over a time horizon.
> >
> >     Note that since our fairness definition is based on Satija et al. (2022), we have not added the formal definition and terminology used in Yin et al. (2023) to avoid confusion.
> >
> > 2. **[More groups and actions]** Our proposed algorithm is naturally extensible to settings with multiple groups or actions, with minimal adjustments:
> >     * Multiple actions: The $\pi$-bisimulation metric, along with our extension, directly supports multiple actions. Therefore, Equation (7) remains valid without any modifications.
> >     * Multiple groups: Equation (7) needs to be estimated for each pair of groups, leading to a total of $\choose{n}{2}$ evaluations, where $n$ is the number of groups.
> >
> > 3. **[High performance of DQN]** Regarding the performance of DQN and PPO, we found that DQN performs comparably to PPO in our experiments, demonstrating the algorithm-agnostic nature of our method. However, we respectfully disagree with the reviewer’s suggestion that this finding contradicts common expectations. This result is typical in relatively simple discrete environments. Even in more challenging discrete scenarios like Atari games, DQN often matches or outperforms PPO. For reference, please see the [CleanRL benchmark on Atari](https://wandb.ai/cleanrl/cleanrl.benchmark/reports/Atari--VmlldzoxMTExNTI), particularly the plots for BeamRider, SpaceInvaders, Pong, and MsPacman. Our implementation is based on CleanRL, as further detailed in Appendix D.
> >
> > 4. **[Comparison to welfare-based RL]** We have appropriately cited the key works on welfare-centric RL in the related work section (Section 6). While welfare-centric fairness offers a promising perspective, its adaptation to machine learning is relatively new compared to group fairness. In this study, we intentionally selected demographic parity as our fairness criterion, given its more established presence in the ML community. This is evidenced by the significantly higher number of citations for demographic parity (Dwork et al., 2012) with 4,483 citations, compared to welfare-centric fairness (Heidari et al., 2018; Cousins, 2021). Nonetheless, we acknowledge that exploring the connection of bisimulation metrics with other fairness notions, including social welfare, is an interesting direction for future work.
> >
> >     If the reviewer is suggesting we include additional baselines from the mentioned papers, we would like to clarify that none of these are directly applicable to our setting:
> >     * Siddique et al. (2020) and Fan et al. (2022) focus on multi-objective RL, whereas our context is single-objective RL.
> >     * Zimmer et al. (2021) addresses multi-agent RL, while our study focuses on single-agent RL.
> >     * Cousins et al. (2022, 2024) are theoretical contributions. Although they include algorithm pseudocode, they lack empirical evaluations, and no existing implementations of these algorithms are available.
> >
> > 5. **[Implementation details]** In our implementation, we first optimize the observation dynamics followed by the reward function. Exploring how this ordering affects performance could be an interesting ablation study to add in the revised version. As mentioned in Line 274 of the original submission, we utilized the One-Plus-One algorithm as our gradient-free optimizer. Furthermore, we have submitted our code, with full implementation details provided in Appendix D.
> > We believe we have addressed all of your concerns. Could you please consider revising the score? If there are still areas that need improvement, we would appreciate any specific feedback on what might be missing.

---

> > > ### Author Response · Authors · 2024-11-19
> > >
> > > ## References
> > > * Yin, Tongxin, et al. "Long-term fairness with unknown dynamics." Advances in Neural Information Processing Systems 36 (2023).
> > > * D'Amour, Alexander, et al. "Fairness is not static: deeper understanding of long term fairness via simulation studies." Proceedings of the 2020 Conference on Fairness, Accountability, and Transparency. 2020.
> > > * Xu, Yuancheng, et al. "Adapting Static Fairness to Sequential Decision-Making: Bias Mitigation Strategies towards Equal Long-term Benefit Rate." Forty-first International Conference on Machine Learning. 2024.
> > > * Deng, Zhihong, et al. "What Hides behind Unfairness? Exploring Dynamics Fairness in Reinforcement Learning." arXiv preprint arXiv:2404.10942 (2024).
> > > * Hu, Yaowei, et al. "Striking a Balance in Fairness for Dynamic Systems Through Reinforcement Learning." 2023 IEEE International Conference on Big Data (BigData). IEEE, 2023.
> > > * Yu, Eric Yang, et al. "Policy Optimization with Advantage Regularization for Long-Term Fairness in Decision Systems." Advances in Neural Information Processing Systems. 2022.
> > > * Dwork, Cynthia, et al. "Fairness through awareness." Proceedings of the 3rd innovations in theoretical computer science conference. 2012.
> > > * Heidari, Hoda, et al. "Fairness behind a veil of ignorance: A welfare analysis for automated decision making." Advances in neural information processing systems 31 (2018).
> > > * Cousins, Cyrus. "An axiomatic theory of provably-fair welfare-centric machine learning." Advances in Neural Information Processing Systems 34 (2021): 16610-16621.
> > > * Siddique, et al. "Learning fair policies in multi-objective (deep) reinforcement learning with average and discounted rewards." International Conference on Machine Learning. PMLR, 2020.
> > > * Fan, Zimeng, et al. "Welfare and fairness in multi-objective reinforcement learning." arXiv preprint arXiv:2212.01382 (2022).
> > > * Zimmer, Matthieu, et al. "Learning fair policies in decentralized cooperative multi-agent reinforcement learning." International Conference on Machine Learning. PMLR, 2021.
> > > * Cousins, Cyrus, et al. "Fair E3: Efficient welfare-centric fair reinforcement learning." 5th Multidisciplinary Conference on Reinforcement Learning and Decision Making. RLDM. 2022.
> > > * Cousins, Cyrus, et al. "On Welfare-Centric Fair Reinforcement Learning." 2024.

---

> > > > ### Author Response · Authors · 2024-11-22
> > > >
> > > > As we near the end of the discussion period, we wanted to follow up to see if you have had a chance to review our rebuttal. If there are any outstanding concerns or questions, we would be happy to address them. Thank you.

---

> > > > > ### Comment · Reviewer_c9Ni · 2024-11-26
> > > > >
> > > > > Thank you for your response. While the authors have addressed most of my concerns, I still recommend conducting additional ablation studies, especially with large group sizes. This would significantly strengthen the paper and further enhance the practical applicability of the proposed algorithm. Accordingly, I have increased my scores to reflect the improvements made.

---

> > > > > > ### Author Response · Authors · 2024-11-26
> > > > > >
> > > > > > Thank you very much for your response and for revising your score. We are glad that the majority of your concerns were addressed. We will include experiments with multiple groups in the final version.

---

> > > > > > > ### Author Response · Authors · 2024-11-28
> > > > > > >
> > > > > > > To address your concern regarding the simplicity of the environments, we have conducted additional experiments with 10 groups (in contrast to the original experiments with 2 groups). These new results demonstrate the scalability of our method to handle a larger number of groups. Please refer to Figure 3 and Table 2 in the revised PDF for details. Due to the rebuttal time constraint, these experiments are run on 5 seeds and compare Bisimulator against ELBERT-PO (the SOTA baseline) and PPO. Full results will be added to the final version.
> > > > > > >
> > > > > > > If your remaining concern regarding the scalability and applicability of our method has been addressed, we kindly request that you consider revising your score.

---

### Official Review · Reviewer_BVZt · 2024-10-30

**Soundness:** 2
**Presentation:** 3
**Contribution:** 3
**Rating:** 6
**Confidence:** 4

**Summary:**

The paper presents a fairness-aware RL approach to satisfy long-term group fairness by capturing behavioral similarities across different groups. The authors show that tuning the observable MDP by minimizing a policy-dependent Bisimulation metric automatically achieves demographic parity. This modified MDP is then leveraged to learn a fair policy using standard RL algorithms.

---------------------------
**Post-rebuttal:**

The rebuttal effectively addressed some of my concerns, and I thus increased my score. However, I still find the scope of the work to be somewhat limited.

**Strengths:**

The paper studies the important problem of long-term fairness, and proposes a novel and somewhat straightforward approach to address it. The authors establish a connection between Bisimulation relation/metric and demographic parity, and exploit this connection for unconstrained optimization of fair policies. Empirical results show better or on par performance of the approach compared to state-of-the-arts. Moreover, the availability of the code supports the reproducibility of these findings (disclaimer: I have not checked the implementation).

**Weaknesses:**

The proposed approach targets a specific class of problems and it is not clear, nor discussed, how it can be generalized/adapted to other problems and settings. The framework relies on a single measure of group fairness which limits its applicability to broader contexts. Particularly that recent literature suggest that individual fairness measures are better suited for fairness-aware learning compared to group-based metrics that are more compliant with statistical analysis.

In addition, it is implicitly assumed that the original MDP is known by the approach which may be a strong requirement (e.g., that R^original(s,a) is given or the transitions are available to sample rollouts). This assumption and how the approach extends to problems with unknown MDPs are not fully discussed in the paper.

The authors claim that modifying the observable MDP occurs outside the RL training loop, so that the problem stays stationary. However, (batch) updates of the dynamics model are still in the main loop, making the policy dependent on the changing MDP, hence, suggesting an overall non-stationary setup.

While the implementation is provided in the supplementary materials, not all parts of the framework are clear. For instance, it would be helpful to discuss how the quantile matching works or whether the expectation in Eq. 7 applies to all possible state and groups or the same state in different groups? i.e., (s,g_i) and (s,g_j)? Also some information about the architecture/etc. would improve clarity.

The main advantage mentioned for modifying the observable MDP over constrained optimization is the ability to use existing RL algorithms. However, it’s not entirely evident how this is a significant benefit, especially given the approach’s limited flexibility in terms of fairness metrics.

Minor:
There are a few references to metrics that are adapted from supervised learning, which implies that metrics are algorithm-dependent. Why this is important?

Typo (line 131):  \tau_a(s’|s,g) should be \tau_a(s’|s,a,g).

The term co-optimization seems to be over-used, e.g., co-optimization of reward function in Sec. 4.1 or co-optimizing J_rew and J_dyn that are learned separately.

line(281-282): the approach is solely based on a single fairness measure, so it is NOT regardless of fairness measure used.

**Questions:**

Is it possible to extend this approach to accommodate different notion of fairness, and if yes, how?

It seems that the Bisimulation metric is defined for discrete state spaces; could you explain if that's the case and how it can be generalized to continuous spaces?

Would it be more intuitive to incorporate the sensitive attribute g as part of the state representation by redefining the state as  S x G, leaving the rest of the MDP and group-conditional definitions unchanged? Would this adjustment change the technical approach?

How the choice of baselines is justified if some of them optimize a different measure of fairness (like EO)? Will the comparisons be fair then?

---

> ### Author Response · Authors · 2024-11-19
>
> Thank you for your review.
>
> ## Weaknesses
> * **[Extension to other problems]** Since our method is based on demographic parity (Dwork et al., 2012), it is in fact applicable to a wide range of problems. We refer the reviewer to (Dwork et al., 2012; Barocas et al., 2023) for detailed discussions on the applicability of demographic parity.
>
> * **[Extension to other fairness notions]** We agree that exploring adaptations to other fairness notions would be a promising direction for future work. However, as highlighted in the limitations section (Section 7), this paper focuses specifically on group fairness. We believe that establishing the link between bisimulation metrics and group fairness could serve as a foundation for future extensions. Moreover, a key novelty of our approach lies in its unconstrained optimization framework, which sets it apart from other methods that rely on constrained optimization.
>
> * **[Assumption of known MDP]** It seems there may be some confusion here. To clarify, our method assumes the MDP is **unknown**—the algorithm does not have access to the explicit functional forms of the reward function or transition dynamics. While we do assume that the MDP is accessible for policy rollouts and sample collection, this is a standard assumption throughout the literature on online RL. Our approach uses only the samples provided by the environment, rather than the explicit forms of $R^{original}(s, a)$ or the transition dynamics. In fact, due to the MDP’s unknown structure, we train a dynamics model $\mathcal{T}_\psi$ on transition samples to evaluate the Wasserstein distance between transition probabilities.
>
> * **[Non-stationarity in training]** In the off-policy deep RL setting, non-stationarity is unavoidable, which is why mechanisms like target networks are typically used to enhance stability. We recognize that our approach introduces additional non-stationarity due to changes in the reward function and observation dynamics. However, by conducting these optimizations outside of the policy optimization loop, we reduce the impact of non-stationarity, thereby increasing the stability of our method. We acknowledge that this point was not clearly communicated in the original text and have revised it for better clarity.
>
> * **[Clarifications on the implementation]** Thank you for your suggestion, we have clarified these points in the revised version:
>     * During each batch update, we divide states into four quartiles for each group. We then compare states from corresponding quartiles across groups as outlined in Equation (7). For example, in the lending scenario, this means matching the top 25% of individuals from group 1 (based on their credit) with the top 25% from group 2. This quantile matching is essential because, as shown in Figure 4, the initial credit distributions of the two groups (representing the states) have minimal overlap.
>     * Detailed architecture and implementation specifics are provided in Appendix D. If there are any further questions, we are happy to address them.
>
> * **[Conceptual benefits of our approach]** We propose an alternative to the widely used constrained optimization methods, offering a perspective that aligns more closely with real-world practices. For instance, our algorithm functions like a third-party regulator that sets rules for fair decision-making without directly engaging in policy optimization. An analogy would be governmental regulations on lending (learned via the bisimulator) and a bank's decision-making policies (learned using an unconstrained RL approach). We believe this regulatory perspective is largely unexplored in the literature, and we hope our work provides initial insights in this direction.
>
> * **[Metrics in supervised learning]** We are not entirely sure we understood your question correctly. Could you please provide further clarification?
>
> * **[Minor comments]** Thank you for highlighting the overuse of the term "co-optimization." We have revised it. Regarding Line 131, this is not a typo as we have used the subscript $a$ in the $tau_a(s’|s, g)$ to represent the action. Regarding Lines 281-282, our point was that the benchmark’s  fairness metric does not affect our method's applicability, regardless of the fact that we use demographic parity. For example, in the lending environment, the metrics are recall, credit gap, and total loans. Unlike our method, some baselines (e.a., A-PPO and EO) explicitly constrain the optimization to achieve equalized recall values, which are directly measured by the benchmark.

---

> > ### Author Response · Authors · 2024-11-19
> >
> > ## Questions
> > 1. **[Extension to other fairness notions]** Could you clarify which specific fairness notion you are referring to? As stated in the limitations section (Section 7), our paper primarily focuses on group fairness.
> >
> > 2. **[Extension to continuous states/actions]** Theoretical extensions of bisimulation metrics to continuous states and actions have already been established (Ferns et al., 2011). Empirically, these metrics have been applied successfully to various continuous control tasks (Zhang et al., 2020). Given this background, extending our theoretical results to continuous states/actions is straightforward. However, as noted in the limitations section, our experimental scope was constrained by the availability of suitable benchmarks.
> >
> > 3. **[Including group membership in the state]** Yes, incorporating group membership into the state representation is feasible and would require minimal adjustments to our method. We opted not to include it for clarity purposes. However, it's worth noting that in our proofs for Theorems 1 and 2, we do include group membership within an augmented state space.
> >
> > 4. **[Choice of the baselines]** The baselines, A-PPO, ELBERT-PO, Lag-PPO, are recent methods in fair RL, with ELBERT-PO being state-of-the-art to the best of our knowledge. It is true that some baselines, like A-PPO and EO, explicitly optimize for equalized recall, whereas our approach targets demographic parity. Given that the benchmark measures recall, this comparison may actually be unfair to our method since it is not designed to directly optimize for equalized recall. Despite this, as shown in our results, our approach outperforms both A-PPO and EO and performs comparably or better than ELBERT-PO.
> >
> > We believe we have addressed all of your concerns. Could you please consider revising your score? If there are still aspects that need improvement, could you clarify what remains unaddressed?

---

> > > ### Author Response · Authors · 2024-11-19
> > >
> > > ## References
> > > * Dwork, Cynthia, et al. "Fairness through awareness." Proceedings of the 3rd innovations in theoretical computer science conference. 2012.
> > > * Barocas, Solon, Moritz Hardt, and Arvind Narayanan. Fairness and machine learning: Limitations and opportunities. MIT press, 2023.
> > > * Ferns, Norm, Prakash Panangaden, and Doina Precup. "Bisimulation metrics for continuous Markov decision processes." SIAM Journal on Computing 40.6 (2011): 1662-1714.
> > > * Zhang, Amy, et al. "Learning Invariant Representations for Reinforcement Learning without Reconstruction." International Conference on Learning Representations. 2020.

---

> > ### Comment · Reviewer_BVZt · 2024-11-19
> >
> > Thanks for the rebuttal and your answers.
> >
> > **Regarding the MDP:** now that you've clarified that the MDP is not actually known and you perform online exploration, wouldn't this limit the applicability of your approach in real-world scenarios?
> >
> > **About the non-stationarity:** I'm still somewhat unconvinced about whether the policy will truly learn effectively if the problem keeps changing (I have read your explanation about convergence to Reviewer Yfvq).

---

> > > ### Author Response · Authors · 2024-11-19
> > >
> > > Thank you for your reply.
> > >
> > > * **MDP:** Our method inherits the limitations common to **any** standard online RL algorithms in its need for an online environment. While offline RL avoids the requirement for online exploration, it comes with its own challenges, such as distribution shift and bias toward the dataset, which are particularly important given the objectives of this work. Furthermore, online RL has demonstrated promising real-world applications, including robotics, fusion reactor control (Tracey et al., 2024), balloon navigation (Bellemare et al., 2020), and data center cooling (Lazic et al., 2018).
> > >
> > >     It is also worth noting that **nearly all** prior works in this area (e.g., Xu et al., 2024; Deng et al., 2024; Hu et al., 2023; Yu et al., 2022; Yin et al., 2023; Satija et al., 2022) focus on online RL. To the best of our knowledge, no existing research has explored offline RL in the context of fair RL, likely reflecting the underexplored nature of this domain.
> > >
> > > * **Convergence guarantees:** As also stated in our response to reviewer Yfvq, providing convergence guarantees in the deep RL setting is virtually impossible.  However, we emphasize that the consistent high performance of our method across 10 seeds provides strong empirical evidence of the convergence of the optimization procedure.
> > >
> > > ## References
> > > * Tracey, Brendan D., et al. "Towards practical reinforcement learning for tokamak magnetic control." Fusion Engineering and Design 200 (2024).
> > > * Bellemare, Marc G., et al. "Autonomous navigation of stratospheric balloons using reinforcement learning." Nature 588.7836 (2020).
> > > * Lazic, Nevena, et al. "Data center cooling using model-predictive control." Advances in Neural Information Processing Systems 31 (2018).
> > > * Xu, Yuancheng, et al. "Adapting Static Fairness to Sequential Decision-Making: Bias Mitigation Strategies towards Equal Long-term Benefit Rate." Forty-first International Conference on Machine Learning. 2024.
> > > * Deng, Zhihong, et al. "What Hides behind Unfairness? Exploring Dynamics Fairness in Reinforcement Learning." arXiv preprint arXiv:2404.10942 (2024).
> > > * Hu, Yaowei, et al. "Striking a Balance in Fairness for Dynamic Systems Through Reinforcement Learning." 2023 IEEE International Conference on Big Data (BigData). IEEE, 2023.
> > > * Yu, Eric Yang, et al. "Policy Optimization with Advantage Regularization for Long-Term Fairness in Decision Systems." Advances in Neural Information Processing Systems. 2022.
> > > * Yin, Tongxin, et al. "Long-term fairness with unknown dynamics." Advances in Neural Information Processing Systems 36 (2023).
> > > * Satija, Harsh, et al. "Group fairness in reinforcement learning." Transactions on Machine Learning Research (2022).

---

> > > > ### Author Response · Authors · 2024-11-28
> > > >
> > > > We would like to inform you about two updates to the paper, one is specifically regarding your earlier concern about the convergence properties:
> > > >
> > > > * **Convergence:** Due to the utilization of the $\pi$-bisimulation (on-policy bisimulation) metric, establishing such convergence, even in the tabular case, is challenging and is an open research question in the literature, to the best of our knowledge. This limitation is **not** due to the specifics of our method.  As noted in Section 3.2.3 of Kemertas et al. (2021), the policy dependence of the $\pi$-bisimulation metric renders the assumptions required for proving policy improvement in DBC (Zhang et al., 2020), and hence convergence to the optimal policy, overly restrictive and seldom satisfied.
> > > >
> > > >     Consequently, convergence proofs for RL methods based on $\pi$-bisimulation metrics are an open topic of research. It requires an intricate analysis on how the fixed-point properties of $\pi$-bisimulation interact with the convergence properties of a bisimulation-dependent policy, as they both rely on one another. This is an interesting research avenue on its own, beyond the primary focus of our paper, which is the application of bisimulation metrics for group fairness. Nonetheless, these methods, including DBC and our approach, have demonstrated strong and consistent empirical performance.
> > > >
> > > >    We have added this theoretical limitation in the revised PDF in the limitations section (Section 7).
> > > >
> > > > * **Additional experiments with several groups:** To address the comments of Yfvq and c9Ni regarding the simplicity of the environments, we have conducted additional experiments with 10 groups (in contrast to the original experiments with 2 groups). These new results demonstrate the scalability of our method to handle a larger number of groups. Please refer to Figure 3 and Table 2 in the revised PDF for details. Due to the rebuttal time constraint, these experiments are run on 5 seeds and compare Bisimulator against ELBERT-PO (the SOTA baseline) and PPO. Full results will be added to the final version.
> > > >
> > > > If there are no remaining concerns, we kindly request that you consider revising your score. Otherwise, we would be happy to address any outstanding issues.

---

> > > > > ### Comment · Reviewer_BVZt · 2024-11-28
> > > > >
> > > > > Thanks for the clarifications and your efforts in revising the manuscript.
> > > > >
> > > > > However, my concern about convergence is not with the combination of Deep RL and the $\pi$-bisimulation metric in general, but specific to your method, where the reward and policy are alternatively optimized. I would assume that constantly changing the MDP requires re-learning the policy as it will become a new problem (going back to the non-stationarity point). While I acknowledge the empirical evidence showing the approach works, it may be challenging to achieve similar performance in other scenarios.
> > > > >
> > > > > Perhaps a key question here is: what if you completely separate the two steps—first optimizing the reward using any exploratory policy, and only updating the policy once the reward optimization is complete?

---

> > > > > > ### Author Response · Authors · 2024-11-29
> > > > > >
> > > > > > Thank you for your reply and your continued engagement during the discussion period.
> > > > > >
> > > > > > That is an interesting suggestion and we have carried out additional experiments to illustrate that our method would work with the modified training loop. We first optimize the reward and/or observation dynamics independently for a certain number of steps without modifying the initially randomized policy. After this phase, we fix the MDP and start the policy optimization, ensuring the stability of policy optimization. While the final performance of Bisimulator remains unaffected, Bisimulator (Reward Only) exhibits a slightly higher recall gap. We will include these experiments as an ablation study in the final version of our work. We believe this training would address your concerns regarding stability as learning the optimal policy is now conducted on a fixed MDP.
> > > > > >
> > > > > > As we are unable to revise the PDF at this stage, we present a table below comparing the new and original variants on the lending scenario with **10** groups. The corresponding learning curves can be accessed via this [anonymous link](https://send.cm/s/6N3T/ICLR_2025_5067).
> > > > > >
> > > > > >
> > > > > > |                                              | Average Return          | Recall Avg          | Recall Std         | Recall Gap          |
> > > > > > |----------------------------------------------|--------------------------|---------------------|--------------------|---------------------|
> > > > > > | PPO + Bisimulator                            | **3918.87 ± 67.58**     | **1.00 ± 0.00**     | 0.00 ± 0.00        | **0.00 ± 0.00**     |
> > > > > > | PPO + Bisimulator (Reward only)              | **3872.32 ± 86.35**     | **1.00 ± 0.00**     | 0.00 ± 0.00        | **0.00 ± 0.00**     |
> > > > > > | PPO + Bisimulator (Stop Bisim Update)        | **3906.93 ± 39.52**     | **1.00 ± 0.00**     | 0.00 ± 0.00        | **0.00 ± 0.00**     |
> > > > > > | PPO + Bisimulator (Reward Only) (Stop Bisim Update) | **3802.05 ± 148.66** | **0.98 ± 0.04**     | **0.02 ± 0.05**    | 0.06 ± 0.12         |

---

### Official Review · Reviewer_3LCW · 2024-11-04

**Soundness:** 2
**Presentation:** 3
**Contribution:** 2
**Rating:** 5
**Confidence:** 3

**Summary:**

This work studies the problem of enforcing long-term fairness in reinforcement learning (RL). Unlike constrained optimization approaches, the authors propose an unconstrained policy optimization algorithm. This algorithm is inspired by the connection between bisimulation metrics and long-term DP fairness in RL. The authors analyze the proposed group-conditioned bisimulation metric and propose a practical algorithm to adjust the reward and observation dynamics to achieve long-term fairness. The effectiveness of the proposed algorithm is demonstrated through extensive numerical studies on two real-world case studies, comparing it with several baselines.

**Strengths:**

1.	The connection between bisimulation and long-term fairness in RL is a novel and insightful approach.
2.	The authors preform thorough numerical studies on two real-world lending and college admission scenarios, comparing their method against up-to-date baselines.

**Weaknesses:**

1.	The clarity of the writing could be improved by providing more details and explanations for certain aspects of the proposed method. See Question section.

**Questions:**

1.	In Definition 5 of DP fairness in RL, the difference inexpected returns is considered for the same state $s$ and different group variables $g_i, g_j$, while in Equation (6), the difference is considered for different state-group pairs $(s_i,g_i)$ and $(s_j,g_j)$. Could you elaborate on the relationship between these two equations?
2.	Please clarify whether the proposed method is applicable to offline or online RL settings. In Algorithm 1, is the dataset $\mathcal{D}$ collected through environment interaction or by using a bisimulator?
3.	Could you explain the role of $\omega$ in Line 12 of Algorithm 1?
4.	The authors could consider mentioning potential limitations of the proposed method.

---

> ### Author Response · Authors · 2024-11-19
>
> Thank you for your review.
>
> ## Weaknesses
> 1. **[Clarifications]** Thank you for your suggestion, we have improved the clarity of the paper in the revised version. Please find the updated document attached, with the changes highlighted in purple text.
>
> ## Questions
> 1.  **[Fixed state vs different state-group pairs]** A bisimulation relation compares all state-group pairs (Castro, 2020), which will include the case where the state is the same, but the group is different. Although it makes sense to have a fixed state, it is highly unlikely to have exact equivalence in practice especially when the state space is large. For example, in the credit score scenario, we might have two individuals belonging to two different groups, but have their credit score differ by 1 point. Such a pair would not be considered for the policy update if we fix the state, but it will be accounted for in the group $\pi$-bisimulation update since they will be considered similar enough. Therefore, we choose to not constrain the bisimulation definition by fixing the state and instead use quantile matching to select the states for comparison.  For example, in the lending scenario, this means matching the top 25% of individuals from group 1 (based on their credit) with the top 25% from group 2. This quantile matching is essential because, as shown in Figure 4, the initial credit distributions of the two groups (representing the states) have minimal overlap.
>
> 2. **[Nature of algorithm (online/offline)]** The proposed algorithm is currently tailored for online RL algorithms and the dataset $\mathcal{D}$ is collected through interaction with the environment. Exploring extensions to offline RL algorithms is a promising direction for future research.
>
> 3. **[Role of $\omega$]** $\omega$ represents the adjustable parameters of the modifiable MDP observation dynamics. Its role is discussed in Lines 249-254 in the initial submission, with additional clarifications provided in the revised version. The use of $\omega$ is context-specific and is explained individually for each environment in Sections 5.1 and 5.2.
>
>     * In the lending environment, $\omega$ represents the credit changes that depend on the applicant’s group membership. For instance, applicants from the disadvantaged group may receive a higher credit increase upon loan repayment, compared to those who belong to the advantaged group. This is a realistic assumption since in practice, banks or other regulators are allowed to override credit scores during their decision making process (Lines 306-312 in the initial submission).
>
>     * In the college admission environment, $\omega$ represents the group specific costs for score modification. These adjustments can be seen as subsidized education for disadvantaged groups, which is a common practice (Lines 418-420 in the initial submission).
>
> 4. **[Limitations]** Thank you for your suggestion. We have mentioned some of the limitations in Section 7, noting the constrained experimental setup due to the lack of a more complicated benchmark in the existing literature. Additionally, our current approach is based on demographic parity fairness. Extending the method to encompass other fairness criteria remains an intriguing avenue for future research.
>
> We believe we have addressed all of your concerns. Since the only weakness you mentioned is on the clarity of the paper, could you please consider raising the score? If not, could you clarify what may still be missing?

---

> > ### Author Response · Authors · 2024-11-22
> >
> > As we near the end of the discussion period, we wanted to follow up to see if you have had a chance to review our rebuttal. If there are any outstanding concerns or questions, we would be happy to address them. Thank you.

---

> ### Author Response · Authors · 2024-11-28
>
> We would like to inform you about additional experiments included in the revised PDF. We have conducted additional experiments with 10 groups (in contrast to the original experiments with 2 groups). These new results demonstrate the scalability of our method to handle a larger number of groups. Please refer to Figure 3 and Table 2 in the revised PDF for details. Due to the rebuttal time constraint, these experiments are run on 5 seeds and compare Bisimulator against ELBERT-PO (the SOTA baseline) and PPO. Full results will be added to the final version.
>
> Additionally, we believe we have addressed all your previous concerns and have improved the clarity of the paper in multiple areas. If you find that all your concerns have been resolved, we kindly request that you consider revising your score. If there are any remaining issues, we would be happy to address them.

---

### Official Review · Reviewer_Yfvq · 2024-11-06

**Soundness:** 2
**Presentation:** 2
**Contribution:** 2
**Rating:** 5
**Confidence:** 3

**Summary:**

The paper studies fairness in reinforcement learning. The author(s) proposed bisimulation metrics to measure long-term fairness. These metrics were further minimized to learn the reward and transition function, based on which the estimated optimal policy is derived. The author(s) further conducted two numerical experiments, including a lending example and a college admission example to investigate the finite sample performance of their proposed method.

**Strengths:**

I summarize the strengths of the paper as follows:

* Although fairness has been widely studied in the machine learning literature, it has been less considered in RL, to my knowledge. In that sense, the problem formulation is relatively "original".
* The approach to ensuring fairness through optimizing bisimulation metrics is "original", to my knowledge. Existing work in RL typically imposes constraints to guarantee fairness.
* The proposed $\pi$-bisimulation-guided reward shaping algorithm is "original", to the best of my knowledge.

**Weaknesses:**

I summarize the weaknesses of the paper as follows:

* One of my major concern lies in the use of the $\pi$-bisimulation-guided approach for ensuring fairness in RL. In particular, the $\pi$-bisimulation metric in (3) is intended to identify equivalent states with same rewards and transition functions. Similarly, the revised $\pi$-bisimulation metric in (7) is intended to identify equivalent state-group pairs with same rewards and transition functions. In other words, if two groups have similar rewards and transition functions, imposing such a constraint is unnecessary as they inherently would achieve similar rewards and transition functions. Conversely, if the rewards and transitions differ between two groups, enforcing such a constraint to align them might result in approximations that differ substantially from their true values. Are there scenarios where deliberately modifying the reward/transition functions could lead to fairer outcomes without significantly compromising performance.

* Definition 5 considers a fixed s between two groups. However, the proposed metrics optimize over different state-group pairs. It makes more sense to minimize across different groups while holding the states constant. Please clarify your reasoning for optimizing over different state-group pairs rather than fixing the state across groups.

* Theorems 1 and 2 appear to be direct extensions of existing results. Theorem 3, on the other hand, is ambiguous. Specifically, Definition 5 involves a constant $\epsilon$. Please clarify which $\epsilon$ value would be attained by minimizing the proposed metric.

* The presentation needs to be enhanced. At several places, it remains unclear to me how the proposed methodology is indeed implemented. For instance, on Page 4, the author(s) mentioned their proposal is to minimize Equation (7). However, it remains unclear what are the parameters being optimized. Similarly, in Algorithm 1, it remains unclear what is the definition of the parameter $\omega$ being optimized on Line 12 (see the Questions Section).

* Current theories did not fully support the validity of the proposal. For instance, it can be seen from Algorithm 1 that the proposed algorithm is iterative in nature, which alternates between estimation of the reward and transition functions and learning of the optimal policy. Please clarify whether such an iterative algorithm would converge.

* The numerical example, particularly the college admission, appears superficial. Can you consider more realistic examples to more effectively evaluate the various algorithms?

* Should $j^{\pi}$ in Definition 5 be $V^{\pi}$? This seems a typo.

**Questions:**

1. What does the "long-term fairness" mean? Why do you want to emphasize "long-term" in the title?
2. What parameters are being optimized when minimizing Equation (7)?
3. What is the definition and role of parameter $\omega$ in Algorithm 1?

---

> ### Author Response · Authors · 2024-11-19
>
> Thank you for your review.
>
> ## Weaknesses
> * **[The use of bisimulation metrics]** The bisimulation relation was originally introduced for labeled transition systems, and its original use case was in fact for **comparing two systems** [Milner, 1989]. In RL, bisimulation metrics are psuedometrics that measure the behavioral similarity between states based on transition dynamics and rewards, **without requiring** the states to belong to the same MDP. While most work on bisimulation metrics in RL has focused on representation learning within a single MDP, these metrics have broader applications beyond representation learning. For instance, they have been used for transfer learning between multiple MDPs (Castro et al., 2010), identifying symmetries (Taylor et al., 2008), and mapping between two MDPs (Rezaei-Shoshtari et al., 2022).
>
>     Our approach introduces a novel application of bisimulation metrics: optimizing specific aspects of an MDP (such as reward functions and observation dynamics) using guidance from these metrics. There are no conceptual or theoretical issues to applying bisimulation metrics in this manner.
>
> * **[Scenario illustrating the effect of modifying the reward]** As an example, consider the lending environment where group 2 is disadvantaged. In this scenario, many individuals in group 2 have a low likelihood of repayment, although some are capable of repaying their loans. A greedy agent that solely maximizes the bank's profit learns that in order to maximize returns, it is optimal to disregard the small chance of repayment of those individuals and automatically reject all applicants from group 2. However, by optimizing the reward function guided by bisimulation, an additional reward term is introduced into the problem. This term, once learned by the Bisimulator, incentivizes providing loans to group 2, leading to higher acceptance rates and improved recall compared to the greedy agent. It is worth noting that fairness-aware algorithms often experience some performance trade-offs relative to purely greedy approaches, this is widely recognized in the literature (Barocas et al., 2023).
>
>
> * **[Fixed state vs different state-group pairs]** A bisimulation relation compares all state-group pairs (Castro, 2020), which will include the case where the state is the same, but the group is different. Although it makes sense to have a fixed state, it is highly unlikely to have exact equivalence in practice especially when the state space is large. For example, in the credit score scenario, we might have two individuals belonging to two different groups, but have their credit score differ by 1 point. Such a pair would not be considered for the policy update if we fix the state, but it will be accounted for in the group $\pi$-bisimulation update since they will be considered similar enough. Therefore, we choose to not constrain the bisimulation definition by fixing the state and instead use quantile matching to select the states for comparison.  For example, in the lending scenario, this means matching the top 25% of individuals from group 1 (based on their credit) with the top 25% from group 2. This quantile matching is essential because, as shown in Figure 4, the initial credit distributions of the two groups (representing the states) have minimal overlap.
>
> * **[Clarification for proof of Theorem 3]** $\epsilon \in \mathbb{R}$ is a small positive threshold that defines how close the iteratively computed pi-bisimulation metric $d_{n}$ is to its fixed-point $d_\sim$. Specifically, we bound $||d_n - d_{\sim}|| \leq ϵ$ by utilizing an existing result stating that the sequence ${||d_n - d_{\sim}||}$ is a contraction that assumes a fixed point $d_{\sim}$ by Banach fixed-point theorem (Castro, 2020). Since we are minimizing Equation 7, updating $d_{n}$ will almost surely reduce to $d_{\sim}$ after $N \geq n$ iterations, which means we would obtain bisimilar state-group pairs where, as was pointed out, the rewards and the transition dynamics would be similar for each group. We combine this fact about bisimulation relations with demographic parity fairness notion such that $\epsilon$ also becomes an acceptable error of how **unfairly** different state-groups are treated in a sequential decision making process.
>
> * **[Implementation details]** Please see our replies to **Questions 2** and **3** below.
>
> * **[Convergence of the algorithm]** We would like to remind the reviewer that our focus is within the deep RL setting, where convergence guarantees are nearly impossible to establish. Therefore, proving convergence for our iterative algorithm in this context is not feasible. Our primary goal is to present a **practical** algorithm with some theoretical foundations based on bisimulation metrics. The empirical results demonstrate the stability and effectiveness of our approach.

---

> ### Author Response · Authors · 2024-11-19
>
> * **[More complex benchmark]** As highlighted in the limitations section (Section 7), the benchmark we utilized in this paper (D'Amour et al., 2020) is currently the **only** established benchmark within the fair RL domain. Several recent works (Xu et al., 2024; Deng et al., 2024; Hu et al., 2023; Yu et al., 2022) have also relied on this same benchmark. While we acknowledge the need for more advanced benchmarks, developing such benchmarks falls outside the scope of our paper. If the reviewer is aware of a more complex benchmark that could enhance our evaluation, we would welcome the suggestion.
>
> * **[Definition 5]** This is not a typo as Definition 5 uses $J^\pi$. We refer the reviewer to Definition 2.2 of Satija et al. (2022).
>
> ## Questions
> 1. **[Definition of long-term fairness]** We provided a brief definition of long-term fairness in Lines 30-31 of the original submission: "ensuring fairness over an extended period rather than at the level of individual decisions." A more formal definition has been added to the revised version, following Yin et al. (2023):
> > Yin et al. (2023) define long-term fairness in RL as the optimization of the cumulative reward subject to a constraint on the cumulative utility, reflecting fairness over a time horizon.
>
>     Note that since our fairness definition is based on Satija et al. (2022), we have not added the formal definition and terminology used in Yin et al. (2023) to avoid confusion.
>
> 2. **[Optimizable parameters]** The optimization procedure for Equation (7) is detailed in Sections 4.1 and 4.2.
>     * In Section 4.1, we describe how we optimize the learnable reward function $R_\phi(s, a, g)$, which is parameterized by a neural network with parameters $\phi$ and optimized using a gradient-based optimizer.
>     * In Section 4.2, we describe how we optimize the modifiable parameters $\omega$ of the observation dynamics. $\omega$ is optimized with a gradient-free optimization method (One-Plus-One). Please see our reply to **Question 3** below for the role of $\omega$.
>
> 3. **[Role of $\omega$]** $\omega$ represents the adjustable parameters of the modifiable MDP observation dynamics. Its role is discussed in Lines 249-254 in the initial submission, with additional clarifications provided in the revised version. The use of $\omega$ is context-specific and is explained individually for each environment in Sections 5.1 and 5.2.
>
>     * In the lending environment, $\omega$ represents the credit changes that depend on the applicant’s group membership. For instance, applicants from the disadvantaged group may receive a higher credit increase upon loan repayment, compared to those who belong to the advantaged group. This is a realistic assumption since in practice, banks or other regulators are allowed to override credit scores during their decision making process (Lines 306-312 in the initial submission).
>
>     * In the college admission environment, $\omega$ represents the group specific costs for score modification. These adjustments can be seen as subsidized education for disadvantaged groups, which is a common practice (Lines 418-420 in the initial submission).
>
> We believe we have addressed all your concerns. Could you consider revising your score? If there are any remaining issues or areas for improvement, we would appreciate it if you could clarify what still needs to be addressed.

---

> > ### Author Response · Authors · 2024-11-19
> >
> > ## References
> > * Castro, Pablo, and Doina Precup. "Using bisimulation for policy transfer in MDPs." Proceedings of the AAAI conference on artificial intelligence. Vol. 24. No. 1. 2010.
> > * Pablo Samuel Castro. "Scalable methods for computing state similarity in deterministic markov
> > decision processes." In Proceedings of the AAAI Conference on Artificial Intelligence, volume 34, pp. 10069–10076, 2020.
> > * Taylor, Jonathan, Doina Precup, and Prakash Panagaden. "Bounding performance loss in approximate MDP homomorphisms." Advances in Neural Information Processing Systems 21 (2008).
> > * Milner, Robin. "Communication and Concurrency." (1989).
> > * Rezaei-Shoshtari, Sahand, et al. "Continuous MDP homomorphisms and homomorphic policy gradient." Advances in Neural Information Processing Systems 35 (2022): 20189-20204.
> > * Barocas, Solon, Moritz Hardt, and Arvind Narayanan. Fairness and machine learning: Limitations and opportunities. MIT press, 2023.
> > * D'Amour, Alexander, et al. "Fairness is not static: deeper understanding of long term fairness via simulation studies." Proceedings of the 2020 Conference on Fairness, Accountability, and Transparency. 2020.
> > * Xu, Yuancheng, et al. "Adapting Static Fairness to Sequential Decision-Making: Bias Mitigation Strategies towards Equal Long-term Benefit Rate." Forty-first International Conference on Machine Learning. 2024.
> > * Deng, Zhihong, et al. "What Hides behind Unfairness? Exploring Dynamics Fairness in Reinforcement Learning." arXiv preprint arXiv:2404.10942 (2024).
> > * Hu, Yaowei, et al. "Striking a Balance in Fairness for Dynamic Systems Through Reinforcement Learning." 2023 IEEE International Conference on Big Data (BigData). IEEE, 2023.
> > * Yu, Eric Yang, et al. "Policy Optimization with Advantage Regularization for Long-Term Fairness in Decision Systems." Advances in Neural Information Processing Systems. 2022.
> > * Satija, Harsh, et al. "Group fairness in reinforcement learning." Transactions on Machine Learning Research (2022).
> > * Yin, Tongxin, et al. "Long-term fairness with unknown dynamics." Advances in Neural Information Processing Systems 36 (2023).

---

> > > ### Author Response · Authors · 2024-11-22
> > >
> > > As we near the end of the discussion period, we wanted to follow up to see if you have had a chance to review our rebuttal. If there are any outstanding concerns or questions, we would be happy to address them. Thank you.

---

> > > > ### Comment · Reviewer_Yfvq · 2024-11-26
> > > > **Post-rebuttal comments**
> > > >
> > > > I would like to thank the author(s) for the responses. I have increased my score to 5, but I still have concerns regarding the proposed methodology. Specifically, in response to the fixed versus different state scenario, the examples mentioned involve two scores, $s_0$ and $s_1$, which differ by only 1. When estimating the value function using function approximation, it is likely that observations from both states will influence the value function at both $s_0$ and $s_1$, not exclusively one or the other. Consequently, such pairs are considered during the policy update due to function approximation, even if we fix the state.
> > > >
> > > > Furthermore, while I understand the challenges of studying convergence in deep RL, I remain uncertain about whether the algorithm converges even in the simplest, tabular setting. I am concerned that the estimator may not converge to the correct target.

---

> ### Author Response · Authors · 2024-11-28
>
> We thank the reviewer for their reply and for revising their score.
>
> * **Fixed versus different states:** We acknowledge that, under function approximation, such pairs could eventually be updated, though this may not always occur. By employing quantile matching, we ensure that similar states are consistently updated. This approach provides a clearer learning signal for the bisimulator, improving its effectiveness in practice.
>
> * **Convergence:** Due to the utilization of the $\pi$-bisimulation (on-policy bisimulation) metric, establishing such convergence, even in the tabular case, is challenging and is an open research question in the literature, to the best of our knowledge. This limitation is **not** due to the specifics of our method.  As noted in Section 3.2.3 of Kemertas et al. (2021), the policy dependence of the $\pi$-bisimulation metric renders the assumptions required for proving policy improvement in DBC (Zhang et al., 2020), and hence convergence to the optimal policy, overly restrictive and seldom satisfied.
>
>     Consequently, convergence proofs for RL methods based on $\pi$-bisimulation metrics are an open topic of research. It requires an intricate analysis on how the fixed-point properties of $\pi$-bisimulation interact with the convergence properties of a bisimulation-dependent policy, as they both rely on one another. This is an interesting research avenue on its own, beyond the primary focus of our paper, which is the application of bisimulation metrics for group fairness. Nonetheless, these methods, including DBC and our approach, have demonstrated strong and consistent empirical performance.
>
>    We have added this theoretical limitation in the revised PDF in the limitations section (Section 7).
>
> * **Additional experiments with several groups:** To address your earlier comment regarding the simplicity of the environments, we have conducted additional experiments with 10 groups (in contrast to the original experiments with 2 groups). These new results demonstrate the scalability of our method to handle a larger number of groups. Please refer to Figure 3 and Table 2 in the revised PDF for details. Due to the rebuttal time constraint, these experiments are run on 5 seeds and compare Bisimulator against ELBERT-PO (the SOTA baseline) and PPO. Full results will be added to the final version.
>
>
> We hope that the new results and our explanation have further addressed your concerns. In that case, we kindly request that you consider revising your score. Otherwise, we would be happy to address them.
>
>
> ### References
> * Kemertas, M., & Aumentado-Armstrong, T. (2021). Towards robust bisimulation metric learning. Advances in Neural Information Processing Systems, 34, 4764-4777.
> * Zhang, Amy, et al. (2020) "Learning Invariant Representations for Reinforcement Learning without Reconstruction." International Conference on Learning Representations.

---

> ### Public Comment · ~Mete_Kemertas1 · 2025-02-11
> **Re. the convergence of pi-bisimulation and policy improvement**
>
> I accidentally came across this review and discussion of your paper, and noticed the reference our 2021 work, where we pointed out the limitation of the Zhang et al. (2021) theoretical result. Regarding the convergence of policy improvement algorithms using bisimulation, I wanted to take a moment to encourage you to view a follow-up paper, Kemertas & Jepson (2022), linked below. This paper shows one way to obtain (arguably) a stronger convergence result by analyzing a certain approximate policy iteration algorithm. Perhaps the first of its kind, the paper shows convergence of alternating updates to the bisimulation metric (fixed-point iteration) and a policy (given value functions approximated over a bisimulation-aggregated state space).
>
> Link: https://openreview.net/forum?id=Ii7UeHc0mO

---

### Author Response · Authors · 2024-11-19
**General Response**

We thank all the reviewers for their time and constructive reviews.

We appreciate that the reviewers have acknowledged our approach as novel (Yfvq, 3LCW, BVZt, c9Ni), clear, and well-structured (c9Ni). The connection between bisimulation metrics and group fairness has been recognized as novel (Yfvq, 3LCW), insightful (3LCW), and thorough (c9Ni). Our implementation is described as practical (c9Ni) and straightforward, with accompanying code (BVZt). Additionally, the experimental results are recognized for their thoroughness (3LCW) and for effectively demonstrating the method's performance against state-of-the-art baselines (c9Ni, BVZt, 3LCW).

## Responses to Common Concerns/Questions
* **Clarity and organization (Yfvq, 3LCW, BVZt, c9Ni):** We have revised the paper for improved clarity and organization. The changes are highlighted in purple. Further edits and refinements will be incorporated into the final version.

* **Simple environments (c9Ni, Yfvq):**  As noted in the limitations section (Section 7), the benchmark we used (D'Amour et al., 2020) is currently the **only** established benchmark within the fair RL domain. Several recent works (Xu et al., 2024; Deng et al., 2024; Hu et al., 2023; Yu et al., 2022) have also relied on this same benchmark. While we agree that more advanced benchmarks would be beneficial, developing them is beyond the scope of our current paper. We would appreciate any suggestions for a more complex benchmark that could further enhance our evaluation.

* **Extension to other fairness definitions (BVZt, c9Ni):** There are over 20 recognized fairness definitions in machine learning (Mehrabi et al., 2021), with some being more commonly adopted. Most works in this area typically focus on **one** fairness notion to make an algorithmic contribution. As stated in our limitations section, our paper adopts demographic parity (Dwork et al., 2012), a well-established concept in both supervised learning and RL. Extending our approach to other fairness definitions is an intriguing direction but lies beyond the scope of a single paper. Nevertheless, our method’s consistent success across various scenarios and metrics confirms that the demographic parity definition has broad applicability and effectiveness, laying a solid foundation for future research into other fairness notions.

## References
* D'Amour, Alexander, et al. "Fairness is not static: deeper understanding of long term fairness via simulation studies." Proceedings of the 2020 Conference on Fairness, Accountability, and Transparency. 2020.
* Xu, Yuancheng, et al. "Adapting Static Fairness to Sequential Decision-Making: Bias Mitigation Strategies towards Equal Long-term Benefit Rate." Forty-first International Conference on Machine Learning. 2024.
* Deng, Zhihong, et al. "What Hides behind Unfairness? Exploring Dynamics Fairness in Reinforcement Learning." arXiv preprint arXiv:2404.10942 (2024).
* Hu, Yaowei, et al. "Striking a Balance in Fairness for Dynamic Systems Through Reinforcement Learning." 2023 IEEE International Conference on Big Data (BigData). IEEE, 2023.
* Yu, Eric Yang, et al. "Policy Optimization with Advantage Regularization for Long-Term Fairness in Decision Systems." Advances in Neural Information Processing Systems. 2022.
* Mehrabi, Ninareh, et al. "A survey on bias and fairness in machine learning." ACM computing surveys (CSUR) 54.6 (2021): 1-35.
* Dwork, Cynthia, et al. "Fairness through awareness." Proceedings of the 3rd innovations in theoretical computer science conference. 2012.

---

> ### Author Response · Authors · 2024-11-28
> **Updates to the Revised PDF**
>
> We have further made the following changes to the revised PDF, currently available on openreview:
> 1. **Additional experiments (Figure 3 and Table 2):** New experimental results with 10 groups, to showcase the scalability of our method to more complex tasks in response to **Yfvq** and **c9Ni**. Please note that due to the time constraint of the rebuttal, these results are obtained on 5 seeds and compare Bisimulator against PPO and ELBERT-PO (the SOTA baseline). Full results will be added to the final version.
> 2. **Convergence:** Addressing the convergence properties of our method in the limitations section (Section 7) in response to **Yfvq** and **BVZt**.

---

### Meta-Review · Area_Chair_8SBa · 2024-12-17

**Metareview:**

This paper proposes using bisimulation methods for design of RL algorithms with group fairness guarantees. Overall, the reviewers found the problem formulation novel and interesting. However, the reviewers had many concerns which were not fully resolved in the rebuttals. For one, reviewers  felt that the algorithm may have convergence issues, and were not sufficiently many theoretical statements to allay these concerns. In the absence of such guarantees, authors concerns remains due to a perceived limited range of experiments. Lastly, authors felt that some of the problem formulation was unclear. From my own personal reading, the authors could do a better part to formally define the mathematical parameter $\omega$ and explain the role it plays in the problem formulation (since $\omega$ seems to be crucial to the entire method). Observation dynamics is not standard, and so while this is a potentially wonderful source of novelty, its current presentation leaves it obscure. Ultimately, the reviewers were left with a lukewarm impression. And given the strength of ICLR as a venue, I err on the side of rejection and encourage the authors to improve this promising manuscript for future submission to an equally strong conference.

**Additional Comments On Reviewer Discussion:**

Unfortunately, and despite my best efforts,  I was unable reviewers to convince reviewers to participate in the discussion. My assessment is therefore based on careful reading of initial and revised reviews, as well as rebuttals, and of my own reading of the submission. While I regret the absence of a robust reviewer discussion, I am confident in my assessment that no single reviewer felt strongly about acceptance, and all reviewers had strong reservations.

---

### Decision · Program_Chairs · 2025-01-22

Reject